**COMMUNICATIONS**

# Offspring born to influenza A virus infected pregnant mice have increased susceptibility to viral and bacterial infections in early life

Henning Jacobsen[1], Kerstin Walendy-Gnirß[1], Nilgün Tekin-Bubenheim[1], Nancy Mounogou Kouassi[1],
Isabel Ben-Batalla[2,3], Nikolaus Berenbrok[2,3], Martin Wolff [4], Vinicius Pinho dos Reis[1], Martin Zickler[1],
Lucas Scholl[1], Annette Gries[1], Hanna Jania[1], Andreas Kloetgen [5], Arne Düsedau[6], Gundula Pilnitz-Stolze[7],
Aicha Jeridi[8,9], Ali Önder Yildirim [8,9], Helmut Fuchs [10], Valerie Gailus-Durner [10], Claudia Stoeger[10],
Martin Hrabe de Angelis [10,11,12], Tatjana Manuylova[13], Karin Klingel[13], Fiona J. Culley [14],
Jochen Behrends [15], Sonja Loges[2,16,17], Bianca Schneider[18], Susanne Krauss-Etschmann[4,19],
Peter Openshaw [14] & Gülsah Gabriel [1,20,21✉]

Influenza during pregnancy can affect the health of offspring in later life, among which neurocognitive disorders are among the best described. Here, we investigate whether maternal influenza infection has adverse effects on immune responses in offspring. We establish a two-hit mouse model to study the effect of maternal influenza A virus infection (first hit) on vulnerability of offspring to heterologous infections (second hit) in later life. Offspring born to influenza A virus infected mothers are stunted in growth and more vulnerable to heterologous infections (influenza B virus and MRSA) than those born to PBS- or poly(I:C)-treated mothers. Enhanced vulnerability to infection in neonates is associated with reduced haematopoetic development and immune responses. In particular, alveolar macrophages of offspring exposed to maternal influenza have reduced capacity to clear second hit pathogens. This impaired pathogen clearance is partially reversed by adoptive transfer of alveolar macrophages from healthy offspring born to uninfected dams. These findings suggest that maternal influenza infection may impair immune ontogeny and increase susceptibility to early life infections of offspring.

A full list of author affiliations appears at the end of the paper.

nfluenza A viruses are some of the most prevalent viruses circulating during winter and causing annual epidemics with up to five million severe cases every year[1]. Although most healthy individuals will only experience mild to moderate disease upon influenza virus infection, pregnant women are at increased risk of developing severe complications[2,3]. Allogeneic pregnancy creates a unique immunological situation since the placenta expresses paternal antigens while being in direct contact to the maternal immune system. To tolerate the foreign tissue, the maternal immune system needs to mount many processes of immune adaptation and tolerance[4,5].

These changes collectively orchestrate an anti-inflammatory and tolerogenic environment that allows placental development and suppresses fetal rejection[6]. Importantly, this state of adapted immunity leads to increased vulnerability towards infection that can ultimately create a contradictory demand of maintaining fetal tolerance versus mounting an inflammatory immune response[7]. The interplay of maternal influenza A virus (IAV) infection and pregnancy-related immune adaptation in a mouse allogeneic pregnancy model that mimics clinical findings was reported recently[8].

Importantly, maternal influenza might also be harmful to the unborn child with unforeseeable consequences for offspring's health. It is known that maternal immune activation (MIA) defined by elevated pro-inflammatory cytokine responses (e.g. IL-6, IL-17A) that might be induced by external stressors of various etiology is a major contributor to neurodevelopmental disorders in the offspring[9-13]. Epidemiological studies implicate an increased risk for schizophrenia or autism-like disorders in offspring born to prenatally infected mothers[14,15]. Thus, there is increasing knowledge that this phenomenon is likely not restricted to neurodevelopmental disease. The association of in utero environmental stress and disease outcome in later life is thought to affect later life through fetal programming.

Recent studies strengthen the hypothesis that maternal infection may prime the developing immune system by mechanisms that go beyond the sole transfer of maternal antibodies[16]. In utero exposure to maternal inflammatory markers, such as cytokines and chemokines, has been reported to affect fetal immune cell activation. MIA seems to increase the concentration of pro-inflammatory cytokines such as IL-2, IL-6, and TNF and to delay the development of innate immunity in the offspring[17,18]. There is also evidence that the maturation of the adaptive immune system might be influenced by the preferential development of Th17 cells in MIA offspring[19]. Notably, to induce MIA during pregnancy most studies utilize syngenically mated mice and artificial immune activators[8,20]. In most animal studies, polyriboinosinic-polyribocytidilic acid (poly(I:C)) or lipopolysaccharide (LPS) is used as a MIA inducing agent in order to mimic viral or bacterial infection, respectively. These models using artificial immunogens possess specific advantages and disadvantages but their representative character regarding actual infections remains unclear.

Here, we show using a two-hit model, that offspring born to influenza A virus infected dams (first hit) are more vulnerable to secondary infections (second hit) in their early life. The underlying mechanisms are multi-factorial, including maternal immune activation upon first hit, low birthweight of the offspring, and the failure of alveolar macrophages from offspring to clear second hit pathogens in early life.

## Results

**Mild influenza causes maternal immune activation**. To study the impact of maternal immune activation (MIA), we intranasally infected BALB/c-mated allogeneic pregnant C57BL/6 females at early gestation (E5.5) with a dose of $10^3$ plaque forming units

(PFU) of the pH1N1 2009 influenza A virus (IAV) (Fig. 1a). The allogeneic pregnancy model was described previously to reflect clinical findings in humans[8]. Maternal influenza A virus infection at E5.5 resulted in moderate disease displayed by delayed weight gain and increased morbidity scores during pregnancy compared to PBS-treated dams (Fig. 1b; Supplementary Fig. 1a, and Supplementary Table 1). In parallel, intraperitoneal poly(I:C)-treated (4 mg/kg) dams showed weight loss that was restricted to day 1 post infection (d. p.i.) compared to PBS-treated dams (Fig. 1b). The IAV-infection dose used was also non-lethal in non-pregnant mice (Fig. 1c). However, mortality cannot be directly compared between both models due to different humane endpoints. Maternal IAV replication was detected in the bronchial and alveolar epithelium of the lung with infiltrated mononuclear cells and partially destructed alveolar epithelium on day 3 p.i. (Fig. 1d). Moreover, high virus titers at various time points after infection were detected in line with delayed virus clearance capacity during allogeneic pregnancy[8] (Fig. 1e). Next, we measured a multiplex panel of cytokines and chemokines involved in MIA. In the lungs, only IAV-infected dams showed the most significant cytokine induction unlike PBS or poly(I:C) treated groups (Fig. 1f–h and Supplementary Fig. 1c–i). IL-6, TNF, MCP-1, and IL-1ß levels were significantly increased in the lungs of IAV-infected dams as compared to PBS or poly(I:C)-treated controls (Fig. 1f–h and Supplementary Fig. 1c–i). In the plasma of IAV-infected dams, only TNF levels were significantly increased compared to PBS-treated control dams (Fig. 1i–k and Supplementary Fig. 1j–p). In general, cytokine induction in dams treated intranasally with poly(I:C) was less potent with respect to cytokine levels in the lung and in the plasma compared to dams receiving poly(I:C) intraperitoneally (Supplementary Fig. 1). In all cases, MIA cytokine induction in the lung was highest in IAV-infected dams. Plasma corticosterone and progesterone levels were mostly unaffected in IAV-infected dams, albeit slight increases were detected compared to control dams (Fig. 1l, m). Overall, these results show that respiratory IAV infection mediates strong and prolonged MIA cytokine induction in the lung, particularly of IL-6, TNF, MCP-1, and IL-1ß.

**Early gestational influenza causes low-birth-weight offspring**. To understand how IAV-induced MIA affects pregnancy outcome, fetuses of IAV-infected or poly(I:C)-treated dams were assessed at gestational day 17.5 (E17.5) and at 2 weeks of age (Fig. 2a). Fetuses and offspring from PBS-treated dams were used as controls. Gestational length was unaffected by maternal IAV-infection or poly(I:C)-treatment when compared to control animals (Fig. 2b). Also, litter size as well as sex distribution within the litters did not significantly differ between dams that were infected with IAV or treated with poly(I:C) compared to PBS-treated dams (Fig. 2c, d). Dams that were treated with poly(I:C) or infected with IAV showed increased placental weight when compared to control dams (Fig. 2e). This phenotype has been previously correlated to compensatory mechanisms upon gestational stress[21,22]. In line with these findings, key nutrient genes were upregulated in placentas from MIA positive dams (Fig. 2f). Fetuses from both, poly(I:C)-treated, and IAV-infected dams presented significantly reduced body weight on E17.5 compared to fetuses from control dams (Fig. 2g). Being small for gestational age (SGA) is a well-known risk factor for later life disease of the offspring[23,24]. This reduced body weight phenotype was continuously present in male and female offspring born to IAV-infected dams but not in offspring born to poly(I:C)-treated dams until early adulthood (Fig. 2h, i). Exchange of litter born to IAV-infected mothers and born to PBS-treated mothers immediately after birth revealed, that this effect is not affected by maternal

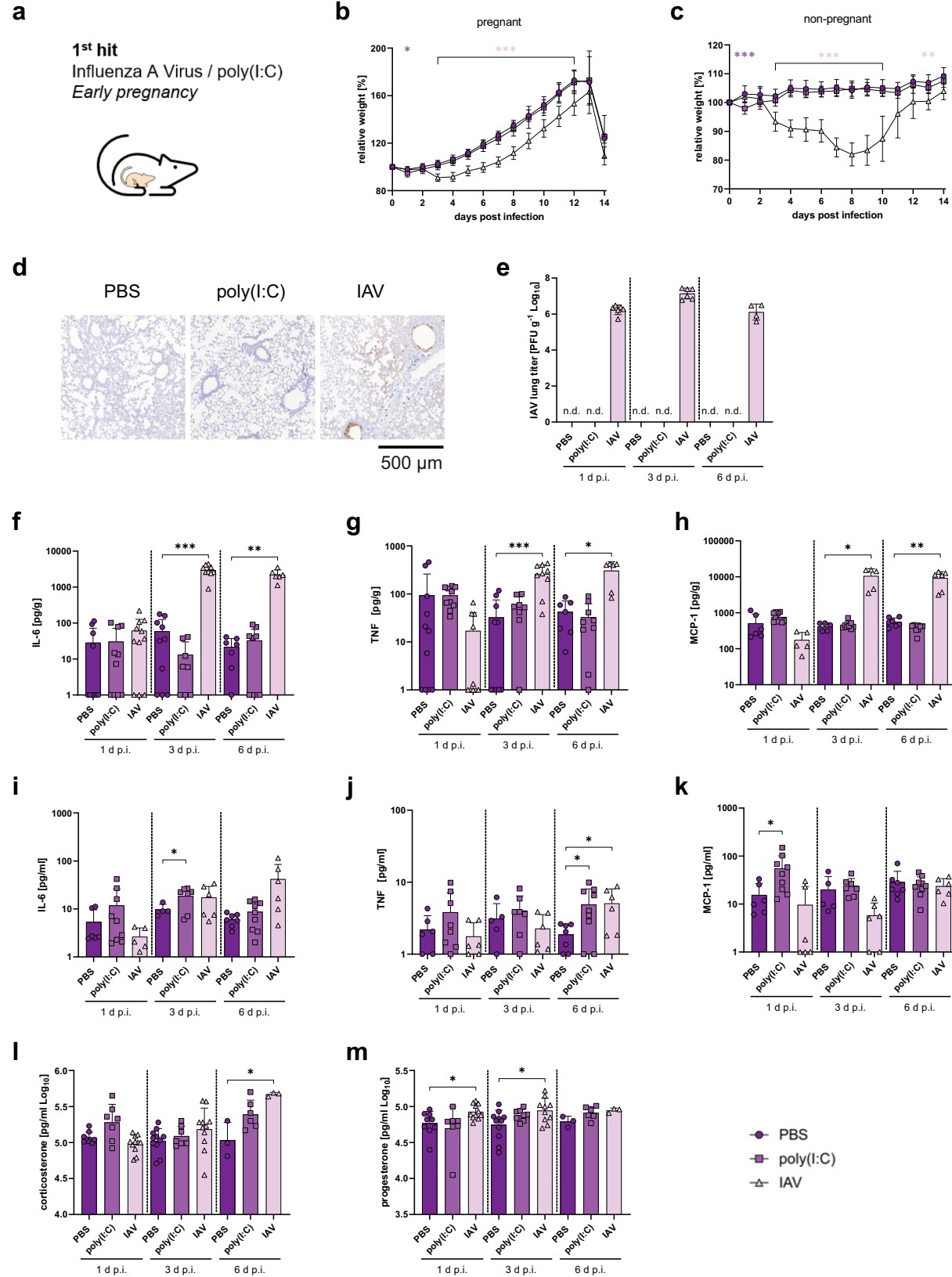

health and brood care in the first two weeks of life (Fig. 2j). No significant differences in lung function were detected in offspring born to IAV or control treated groups at 6 weeks and 20 weeks of age (Supplementary Fig. 2). However, a slight impairment in lung function was observed in IAV-infected male offspring, albeit statistically not significant, compared to male controls.

**Increased susceptibility of offspring towards early life infections.** Challenge experiments with 2- and 6-week-old offspring born to poly(I:C)-treated or IAV-infected dams as well as born to control dams were performed to evaluate functional immunity in this offspring (Fig. 3a). To validate the second hit model, homologous and semi-homologous second hits were performed

**Fig. 1 1st hit maternal influenza virus infection and immune activation. a** The first hit was performed during early pregnancy by application of either poly (I:C) or infection with IAV. **b** Pregnant C57BL/6 mice were treated intraperitoneally (i.p.) with 4 mg/kg poly(I:C) ($n = 24$) or infected intranasally (i.n.) with $10^3$ PFU of IAV ($n = 24$) on day 5.5 of gestation. Phosphate-buffered saline (PBS)-treated (i.n.) dams were used as controls ($n = 20$). Weight development was monitored for 14 days post infection (d p.i.), $p = 0.0003$ (1 d p.i.), $p < 0.0001$ (3-12 d p.i.). **c** Weight of non-pregnant C57BL/6 mice treated with PBS, poly(I:C) or infected with IAV ($n = 10$), $p = 0.0003$ (1 d p.i.), $p < 0.0001$ (3–10 d p.i.), $p = 0.0044$ (13 d p.i.). **d** Representative immunohistology of IAV NP antigen in maternal lungs at 3 days post infection, Scale bar $= 500$ μm. **e** IAV titers as plaque forming units (PFU) per gram were measured in whole lung homogenates from pregnant dams treated with PBS ($n = 4, 5, 6$ [1, 3, 6 d p.i.], poly(I:C) ($n = 6, 4, 6$ [1, 3, 6 d p.i.] or infected with IAV ($n = 6, 6, 4$ [1, 3, 6 d p.i.] by plaque assay. **f–h** Lung cytokines IL-6, $p < 0.0001$ (3 d p.i.), $p = 0.0010$ (6 d p.i.) (**f**); TNF, $p = 0.0001$ (3 d p.i.), $p = 0.0103$ (6 d p.i.) (**g**); and MCP-1, $p = 0.0195$ (3 d p.i.), $p = 0.0058$ (6 d p.i.) (**h**) detected by Luminex assay in lungs of pregnant mice treated with PBS (IL-6, $n = 10, 10, 7$ [1, 3, 6 d p.i.]); (TNF, $n = 11, 10, 7$ [1, 3, 6 d p.i.]); (MCP-1, $n = 6, 5, 7$ [1, 3, 6 d p.i.]), poly(I:C) (IL-6, $n = 10, 8, 9$ [1, 3, 6 d p.i.]); (TNF, $n = 10, 8, 9$ [1, 3, 6 d p.i.]) and (MCP-1, $n = 11, 9, 10$ [1, 3, 6 d p.i.]) or infected with IAV (IL-6, $n = 11, 9, 6$ [1, 3, 6 d p.i.]), (TNF, $n = 10, 9, 6$ [1, 3, 6 d p.i.]); (MCP-1, $n = 5, 5, 6$ [1, 3, 6 d p.i.]) at E5.5. **i–k** Plasma cytokines IL-6, $p = 0.0343$ (**i**); TNF, $p = 0.0218$ poly(I:C), $p = 0.0429$ (**j**) and MCP-1, $p = 0.0329$ poly(I:C) (**k**) measured by Luminex assay in plasma of pregnant mice treated with PBS (IL-6, $n = 6, 4, 7$ [1, 3, 6 d p.i.]); (TNF, $n = 6, 5, 7$ [1, 3, 6 d p.i.]; (MCP-1, $n = 6, 5, 7$ [1, 3, 6 d p.i.]); poly(I:C) (IL-6, $n = 9, 7, 9$ [1, 3, 6 d p.i.]); (TNF, $n = 8, 7, 9$ [1, 3, 6 d p.i.]); (MCP-1, $n = 9, 7, 9$ [1, 3, 6 d p.i.]) or infected with IAV (IL-6, $n = 5, 6, 6$ [1, 3, 6 d p.i.]); (TNF, $n = 6, 6, 6$ [1, 3, 6 d p.i.]); (MCP-1, $n = 6, 6, 6$ [1, 3, 6 d p.i.]) at E5.5. Values are normalized to organ weight if applicable. **l, m** Hormones (corticosterone, $p = 0.0456$ (**l**) and progesterone, $p = 0.0247$ (1 d p.i.), $p = 0.0254$ (3 d p.i.) (**m**) measured by ELISA in plasma of pregnant mice treated with PBS ($n = 9, 11, 3$ [1, 3, 6 d p.i.]; poly(I:C) ($n = 7, 7, 6$ [1, 3, 6 d p.i.] or infected with IAV ($n = 11, 11, 11$ [corticosterone 1, 3, 6 d p.i.], $n = 11, 10, 3$ [progesterone, 1, 3, 6 d p.i.] at E5.5. All data are presented as mean and SD. Different groups are depicted in dark circles (PBS), medium squares (poly(I:C)) or light triangles (IAV) in violet colors. Cytokine levels that were below detection limit were set to the kit's lower detection limit of 1 pg/g. The statistical significance was calculated by Welch's t test (two-tailed) (*$p < 0.05$, **$p < 0.01$, ***$p < 0.001$). PBS-treated groups were used as a reference and compared to IAV-infected groups in all statistical analyses unless stated otherwise. Non-significant comparisons are not depicted within the respective figures. n. d. not detectable. Source data are provided as a Source Data file.

on 2-week-old offspring. Thus, offspring born to PBS-treated or IAV- (H1N1) infected dams were infected with either the same H1N1 virus (homologous) or a H3N2 reassortant (WSN backbone, semi-homologous) at 2 weeks of age. As expected, juvenile offspring presented full protection against homologous challenge and partial protection against semi-homologous protection (Supplementary Fig. 3), likely due to perinatal transfer of protective antibodies confirming the validity of the preclinical two-hit animal model. Heterologous second hits were performed using influenza B virus (IBV) or Methicillin-resistant *Staphylococcus aureus* (MRSA) as a viral or bacterial hit, respectively. Upon infection with $10^5$ PFU of IBV in 2-week-old offspring, male but not female offspring born to IAV-infected mothers present increased lethality compared to offspring born to poly(I:C)- or PBS-treated dams (Fig. 3b, c). In line, infection with $10^8$ Colony Forming Units (CFU) of MRSA resulted in increased lethality in 2-week-old offspring born to IAV-infected dams (Fig. 3d). Because these animals died before sex determination was possible, we were not able to detect potential sex-specific effects for MRSA-infected, juvenile offspring. Cytokine response in offspring after second hit was comparable between males and females. Thus, data are shown without sex-stratification for clarity. In general, cytokine responses were also comparable across groups with the exception of increased IL-2 levels in IAV-offspring after IBV challenge (Fig. 3g). Infection of 6-week-old offspring did not reveal major differences in lethality, cytokine expression or lung histology across groups (Supplementary Figs. 4 and 5), suggesting a window of vulnerability particularly 2 weeks after birth. Viral and bacterial clearance was delayed in 2-week-old offspring born to IAV-infected dams compared to offspring born to PBS-treated dams (Fig. 3k, l and Supplementary Fig. 6). These data show that offspring born to IAV-infected dams are more vulnerable towards secondary viral and bacterial infections in early life.

**Early gestational influenza affects offspring's hematopoiesis.** As we observed increased disease vulnerability in offspring born to IAV-infected mothers, we assessed the frequencies of hematopoietic stem and progenitor cells (Fig. 4a) in the bone marrow of male and female 2- and 6-week-old offspring born to early gestational PBS-treated, poly(I:C)-treated or IAV-infected dams by flow cytometry. Sex was not identified as a potential

confounder for the measurement of relative cell population frequencies and therefore respective data sets are shown without sex-stratification for clarity. The frequencies of hematopoietic stem cells and progenitor cells show a tendency to be increased, particularly in 2-week-old offspring born to IAV-infected mice, compared to offspring born to PBS-treated mice (Fig. 4b, c and Supplementary Fig. 7b, c). Assessment of common myeloid progenitors (CMPs) and granulocyte/macrophage progenitors (GMPs) in juvenile offspring, showed a trend towards increased frequencies in offspring born to IAV-infected mothers, respectively (Fig. 4d, e). Increased CMP frequencies remained in 6-week-old offspring born to IAV-infected dams (Supplementary Fig. 7d). No difference was observed in the frequencies of myeloid/erythroid progenitors (MEPs) in 2-week and 6-week-old offspring (Fig. 4f; Supplementary Fig. 7f). To provide a first overview about potential consequences, we also analyzed a variety of major immune cell subtypes in the offspring's spleens and lungs. Indeed, B cell frequencies were reduced in lungs but not spleens from offspring born to IAV-infected dams (Fig. 4g, j). Additionally, while NK cell frequencies were enhanced in spleens, they were decreased in lungs from offspring born to IAV-infected mice (Fig. 4h, k). On the other hand, regulatory T cell ($T_{reg}$) frequencies showed an increase in those offspring's lungs but not spleens (Fig. 4i, l). Because alveolar macrophages (AM) play a pivotal role during early stages of infection[25], we also measured AM frequencies in the offspring's lungs. A trend towards an increase of AM frequencies was observed in offspring born to IAV-infected mothers (Fig. 4m and Supplementary Fig. 7m). These data indicate that major immune cell populations important for the clearance of pathogens are altered in the lungs of offspring born to influenza virus infected dams.

**Functionally impaired alveolar macrophages in offspring.** The results described so far support the concept, that early gestational influenza virus infection affects offspring's immune development at various levels. Although only subtle differences in AM frequencies were observed, we further investigated the functional capacity of these AMs since they are known to play a crucial role in innate immunity and pathogen clearance[26]. Thus, we performed an intranasal adoptive transfer of AMs between 2-week-old offspring born to control dams or 2-week-old offspring born

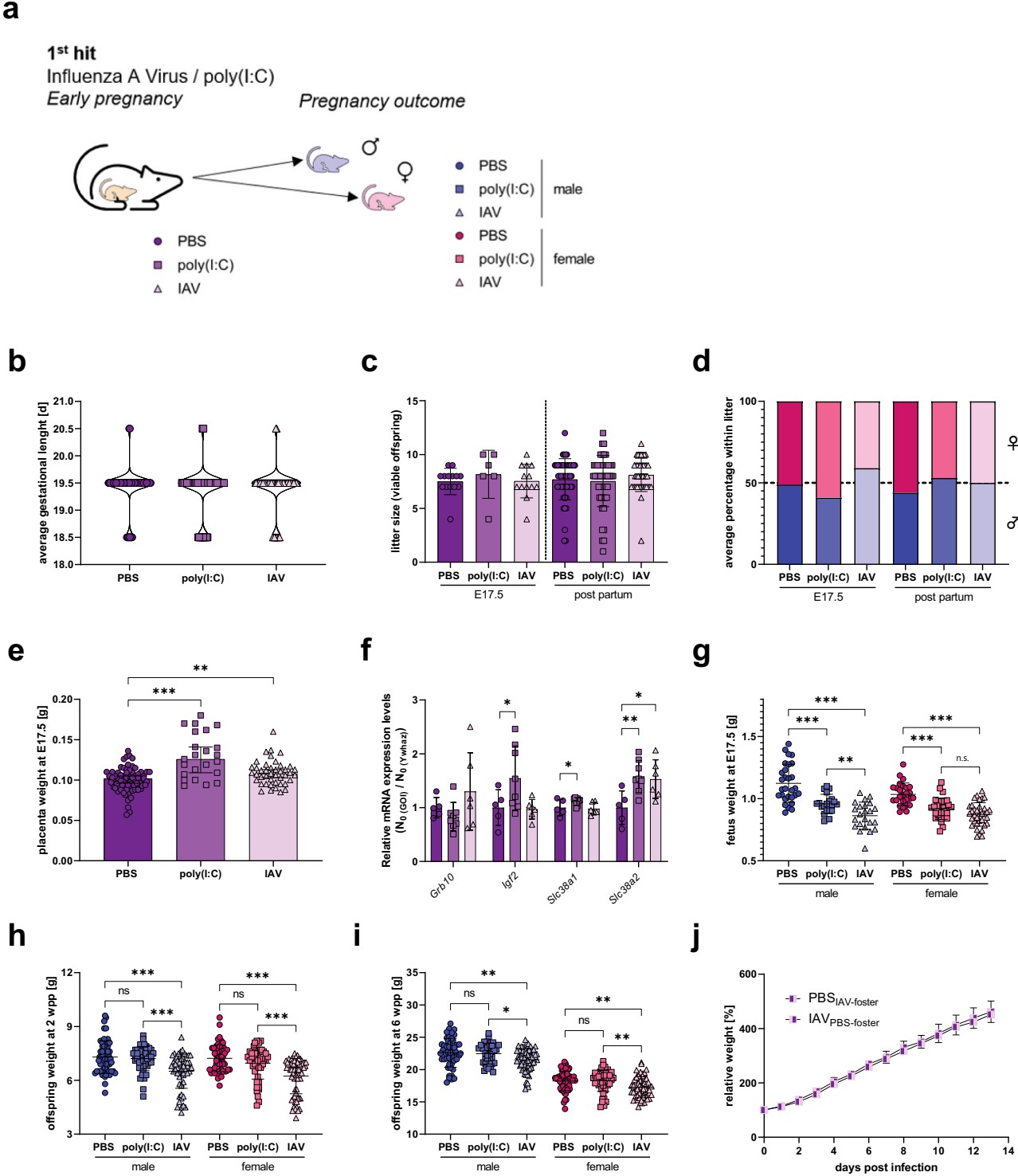

to dams that were either treated with poly(I:C) or infected with IAV. Both, male and female offspring were subsequently infected with IBV to assess the role of transferred AMs in their respective immune response (Fig. 5a). Transfer of AMs from offspring born to poly(I:C)-treated or IAV-infected dams into offspring born to healthy mice did not affect male or female offspring's weight development after infection (Fig. 5b, e). Also, transfer of AMs from offspring born to control dams into offspring born to poly(I: C)-treated dams did not affect immediate disease outcome in males or females (Fig. 5c, f). Importantly, when transferring AMs

from offspring born to healthy mothers into offspring of the same sex born to IAV-infected dams, the offspring showed better recovery compared to pups that did not receive any AMs (Fig. 5d, g). Furthermore, we show that application of AMs derived from MIA offspring into control offspring does not affect viral titers after subsequent IBV challenge as compared to untreated controls (Fig. 5h). Also, the application of control AMs into offspring born to poly(I:C)-treated dams, did not affect virus titers (Fig. 5i, left). In contrast, offspring born to IAV-infected dams that had increased IBV-titers upon heterologous challenge, presented

**Fig. 2 1st hit maternal influenza and reproductive outcome. a** Pregnancy outcome was assessed at embryonic day 17.5 (E17.5) or post partum (pp) with 2-week-old offspring. **b** Gestational length in days (d) of dams that were treated with phosphate-buffered saline (PBS) ($n = 66$), treated with 4 mg/kg poly (I:C) ($n = 65$), or infected with $10^3$ PFU IAV ($n = 61$). **c** Litter size of PBS-treated dams at E17.5 ($n = 14$) and 2 w p.p. ($n = 53$), poly(I:C)-treated dams at E17.5 ($n = 6$) and 2 w p.p. ($n = 48$) and IAV-infected dams at E17.5 ($n = 13$) and 2 w p.p. ($n = 43$). Only viable fetuses were counted. **d** Percentage of male fetuses per litter for PBS-treated dams at E17.5 ($n = 10$) and 2 w p.p. ($n = 53$), poly(I:C)-treated dams at E17.5 ($n = 6$) and 2 w p.p. ($n = 48$) and IAV-infected dams at E17.5 ($n = 13$) and 2 w p.p. ($n = 43$). Only viable fetuses were counted. Fetal sex was determined by PCR, sex of 2-week-old offspring was determined by physiological examination. **e** Placenta weight in grams (g) from dams treated with PBS ($n = 72$), treated with poly(I:C) ($n = 23$, $p < 0.0001$) or infected with IAV ($n = 43$, $p = 0.0062$) at E17.5. **f** Relative mRNA expression of nutrient supply genes of growth inhibitory gene *Grb10* (growth factor receptor bound protein 10); growth stimulatory gene *Igf2* (insulin-like growth factor II), $p = 0.0441$ poly(I:C); *Slc38a1* (solute carrier family 38, member 1), $p = 0.049$ and *Slc38a2* (solute carrier family 38, member 2), $p = 0.0059$ (poly(I:C), $p = 0.0148$ (IAV) of placentas from fetuses from PBS ($n = 5$), poly(I:C) ($n = 8$) or IAV ($n = 6$) infected dams. The relative expression in the PBS group was set to 1 for each gene after normalization to the housekeeping gene *Ywhaz*. **g** Fetuses were weighed on E17.5 for PBS-treated dams ($n = 29$ males, 26 females), poly(I:C)-treated dams ($n = 20$ males, $p < 0.0001$, 29 females; $p < 0.0001$), and IAV-infected dams ($n = 23$ males, $p < 0.0001$, 35 females, $p < 0.0001$). Only viable fetuses were included. **h** Offspring weight at 2 weeks of age (2 weeks post partum; 2 wpp) born to mothers treated with PBS ($n = 54$ males; $n = 56$ females), poly(I:C) ($n = 47$ males, $p = 0.8791$; $n = 49$ females, $p = 0.2665$), and IAV ($n = 59$ males, $p < 0.0001$; $n = 67$ females, $p < 0.0001$). **i** Offspring weight at 6 weeks of age (6 wpp) born to mothers treated with PBS ($n = 48$ males, $n = 57$ females), poly(I:C) ($n = 36$ males, $p = 0.9430$; $n = 46$ females, $p = 0.9831$), and IAV ($n = 69$ males, $p = 0.0064$; $n = 57$ females, $p = 0.039$). **j** Offspring born to PBS-treated mothers was exchanged with offspring born to IAV-infected mothers immediately after birth and weight gain was observed for each animal individually during the first 2 weeks of life. Each group consisted of two litters with eight pups each. All data are presented as mean and SD. Different groups of mothers are depicted in dark circles (PBS), medium squares (Poly(I:C)), or light triangles (IAV) in violet colors. Different groups of male (♂) offspring are depicted in dark circles (PBS), medium squares (Poly(I:C)), or light triangles (IAV) in blue colors and female (♀) offspring in red colors. The statistical significance was calculated by two-tailed ANOVA (*$p < 0.05$, **$p < 0.01$, ***$p < .001$). PBS-treated groups were used as a reference in all statistical analyses. Source data are provided as a Source Data file.

reduced IBV titers after application of AMs derived from PBS-offspring (Fig. 5i, right). These data indicate that early gestational influenza virus infection impairs the functionality of offspring's AMs and this is rescued in part by the transfer of unaffected AMs prior to infection. Analysis of lung cytokine levels revealed that the application of AMs prior to infection with IBV generally increased the expression levels of TNF in uninfected and infected offspring (Supplementary Fig. 8a–d). IL-6 expression in control and infected offspring seems to be unaffected by AM-transfer (Supplementary Fig. 8e–h). Interestingly, transfer of PBS-macrophages into offspring born to MIA-mothers seem to reduce IL-2 expression that was previously shown to be increased in IAV-offspring upon IBV infection (Supplementary Fig. 8i–l and Fig. 3g). Collectively, these results suggest that early gestational influenza virus infection impairs respiratory pathogen clearance by AMs in the offspring.

## Discussion

Maternal immune activation (MIA) is commonly defined as an increase of pro-inflammatory cytokines (e.g. IL-6, TNF, and MCP-1) during pregnancy[13]. In the present study, we used a semi-allogeneic mouse pregnancy model to show that a single-dose application of poly(I:C) on gestational day 5.5 results in a short-lived immune activation early after application. In contrast, infection with IAV results in a stronger and prolonged increase of MIA markers and has an adverse effect on resistance to infection in offspring. As previously shown in this allogeneic pregnancy model, no virus transfer occurs from mother-to-fetus[8]; therefore, the adverse effects observed in offspring are associated with MIA during early gestation.

In our study, we chose doses of IAV and poly(I:C) that lead to mild-to-moderate disease with pronounced MIA. Higher doses of virus or poly(I:C) applied at E 5.5 resulted in maternal death or abortion, the latter especially upon poly(I:C)-treatment at 20 mg/kg. Although MIA is more severe in dams infected with IAV compared to poly(I:C), both show comparably small for gestational age phenotypes in the fetuses. Since low birth weight is a major risk factor for poor health in later life, a comparison of both models in this regard is reasonable and important to evaluate findings from other studies restricted to the use of poly(I:C) or other artificial immunogens[23,24]. Most previously published

models use single-dose application of artificial immunogens, hence we followed this scheme to accurately reflect potential mechanistic differences[27]. We speculate that repetitive rather than single-dose stimulation with poly(I:C) might be a suitable method by which to mimic infection.

We focused our studies on the effects of MIA during early pregnancy because evidence from human clinical studies show that early gestational events are more likely to cause long-term health deficits in the offspring, while stressors during late pregnancy are more likely to cause abortion or maternal death[28,29]. It should be noted that most studies on MIA in mice are carried out in mid or late gestation[20]. This may be because early gestational challenge is technically difficult to achieve since early pregnancy is hard to reliably identify in mice. Common assessment of a copulation plug goes in line with high false-positive pregnancy rates as well as difficulties in reliable palpation before gestational day 10 [30]. Additionally, measurement of pregnancy hormones requires invasive sampling that might influence subsequent experiments. While some work has been done on the effects of timing of MIA on the offspring's neurocognitive development, a detailed understanding on the effects of timing on the offspring's immune system remains elusive[11,28]. In this study, we show that early gestation represents a critical phase of fetal development and that gestational stress results in increased vulnerability of the offspring in the early postnatal period.

Mild maternal influenza or poly(I:C)-application during early gestation did not affect gestational length, litter size, and sex distribution of the offspring. However, increased placental weight and a dysregulation of genes related to transplacental nutrition transfer and metabolism indicated disruption of pregnancy homeostasis, which presents a potential link to runted offspring[21,22,31]. This further highlights the importance of birth weight as a marker for MIA-related effects on the offspring. Importantly, offspring born to poly(I:C)-treated mice recovered the phenotype of relatively reduced body weight within the first 2 weeks of life, whereas offspring born to IAV-infected dams continued to show reduced body weight until early adulthood.

Restricted fetal growth is a significant risk factor for reduced lung function in children[32], especially if postnatal growth is reduced[33]. In our model, 6-week-old male offspring born to IAV-infected mothers displayed a slight reduction in lung function,

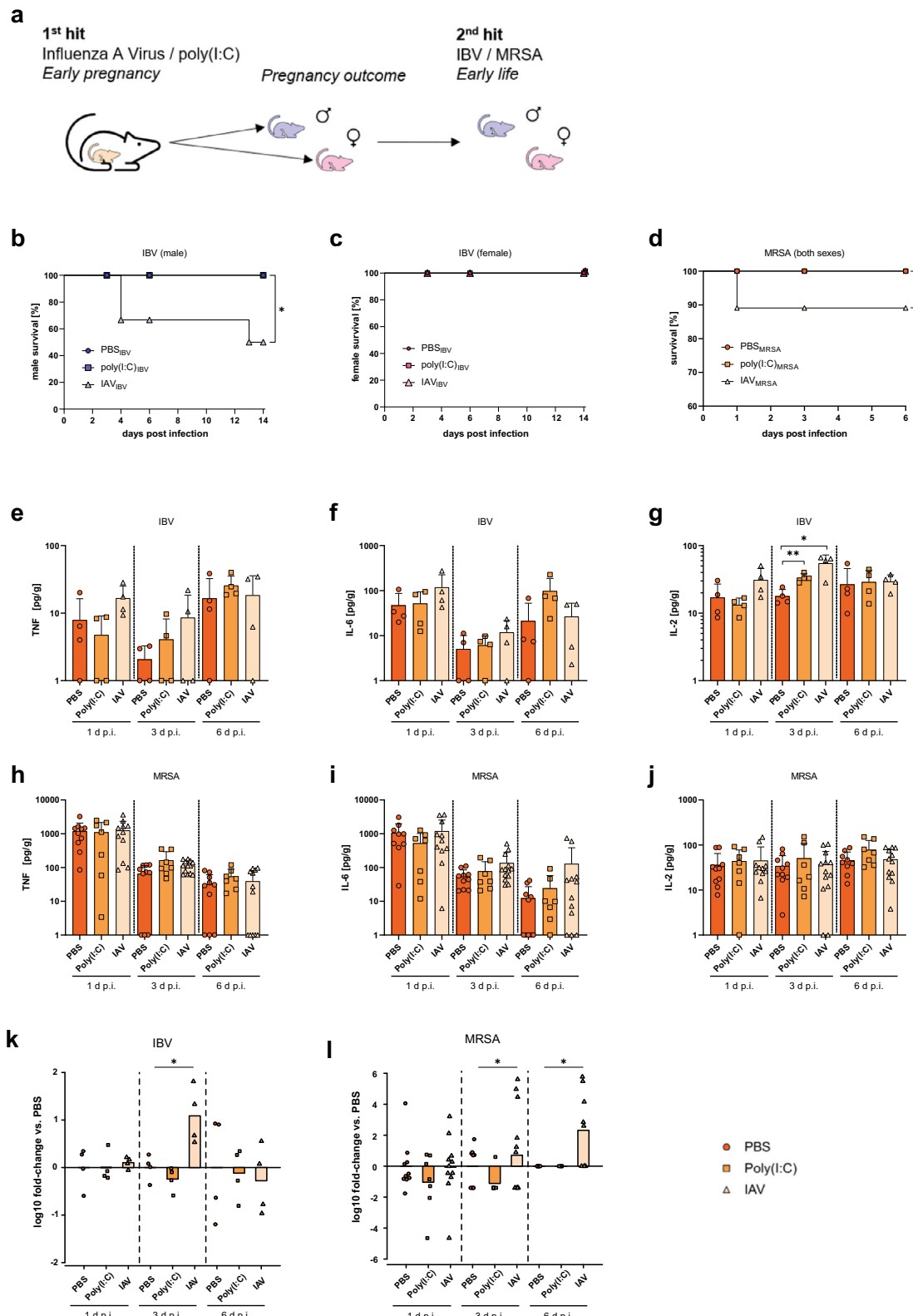

albeit statistically not significant. However, potential delays in lung development were outgrown in all assessed groups at 20 weeks of age.

Infection of juvenile, 2-week-old offspring revealed increased vulnerability towards respiratory infection with both viral and bacterial secondary hits. Notably, upon challenge with IBV, increased vulnerability was restricted to male offspring. This

parallels epidemiological observations in infants born small for gestational age and showing increased vulnerability in later life[34]. Due to ethical and regulatory restrictions, we were not able to detect sex-specificity in the offspring's vulnerability against MRSA. Nevertheless, like the infection with IBV, offspring born to IAV-infected dams showed increased lethality upon MRSA infection, compared to offspring born to PBS- or poly(I:C)-treated dams.

**Fig. 3 2nd hit susceptibility of offspring from influenza infected dams to early life infections. a** Second-hit experiments were performed in 2-week-old offspring using influenza B virus (IBV) or Methicillin-resistant *Staphylococcus aureus* (MRSA) for a respiratory challenge. **b** Survival of 2-week-old male offspring born to IAV-infected dams ($n = 11$ [day 3], $n = 9$ [day 6], $n = 7$ [day 14]) or born to control treated dams with polyinosinic:polycytidylic acid (poly (I:C))-treated ($n = 11$ [day 3], $n = 9$ [day 6], $n = 7$ [day 14]) or PBS ($n = 9$ [day 3], $n = 7$ [day 6], $n = 5$ [day 14]) after infection with $10^5$ plaque forming units (PFU) of IBV; $p = 0.0355$. **c** Survival of 2-week-old female offspring born to IAV-infected dams ($n = 13$ [day 3], $n = 11$ [day 6], $n = 9$ [day 14]) or born to control treated dams with poly(I:C)-treated ($n = 11$ [day 3], $n = 9$ [day 6], $n = 7$ [day 14]) or PBS ($n = 8$ [day 3], $n = 6$ [day 6], $n = 4$ [day 14]) after infection with $10^5$ PFU of IBV. Survival was monitored in three independent experiments. **d** 2-week-old offspring (both sexes) born to poly(I:C)-treated ($n = 21$ [day 1], 14 [day 3], 7 [day 6]) or IAV-infected ($n = 55$ [day 1], 34 [day 3], 16 [day 6]) dams were infected with $10^8$ CFU of MRSA; $p = 0.0167$. Offspring born to PBS-treated dams ($n = 38$ [day 1], 25 [day 3], 12 [day 6]) were used as controls. Survival was monitored on day 1, 3 or 6 p.i. in three independent experiments. **e–g** Cytokines (TNF (**e**), IL-6 (**f**), and IL-2 (**g**)), $p = 0.0034$ poly(I:C), $p = 0.0219$ measured by Luminex assay in lungs of 2-week-old offspring (both sexes) born to PBS- or poly(I:C)-treated or IAV-infected offspring ($n = 4$) after infection with $10^5$ PFU of IBV measured at 1, 3, or 6 d p.i. **h–j** Cytokines (TNF (**h**), IL-6 (**i**), and IL-2 (**j**)) measured by Luminex assay in lungs of 2-week-old offspring (both sexes) born to PBS- ($n = 10$) or poly(I:C)-treated ($n = 7$) or IAV-infected offspring ($n = 11$) after infection with $10^8$ CFU of MRSA measured at 1, 3, or 6 d p.i. **k** IBV lung titer in 2-week-old offspring (both sexes) born to PBS-, poly(I:C)-treated, or IAV-infected ($n = 4$) dams, measured on day 1, 3 ($p = 0.0270$), and 6 p.i. is shown as fold-change in lung titers compared to the respective PBS group of the indicated time point. **l** MRSA lung titer in 2-week-old offspring (both sexes) born to PBS- ($n = 10$ [1 d p.i.], 10 [3 d p.i.], 7 [6 d p.i.]), poly(I:C)-treated ($n = 7$ [1 d p.i.], 7 [3 d p.i.], 4 [6 d p.i.]) or IAV-infected ($n = 11$ [1 d p.i.], 13 [3 d p.i., $p = 0.0189$], 9 [6 d p.i., $p = 0.0223$]) dams, measured on days 1, 3, and 6 p.i. is shown as fold-change in lung titers compared to the respective PBS group of the indicated time point. All $n$ represent number of male and/or female offspring from respective groups. Groups were merged for clarity if no difference between males and females was observed. Data in (**e–j**) are presented as mean and SD. Different groups of male offspring are depicted in dark circles (PBS, medium squares (Poly(I:C)) or light triangles (IAV) in blue colors and female offspring in red colors. Different groups of offspring that were not stratified by sex are depicted in orange colors with the same icons. Cytokine levels that were below detection limit were set to the kit's lower detection limit of 1 pg/g. The statistical significance was calculated by Welch's $t$ test (two-tailed) (**e–j, k, l**) without multiple comparison or Log Rank test (**b–d**) (*$p < 0.05$, **$p < 0.01$). PBS-treated groups were used as a reference and compared to IAV-infected groups in all statistical analyses unless stated otherwise. Source data are provided as a Source Data file.

These data clearly show that early gestational influenza, but not poly(I:C)-induced MIA, affects functional immunity towards respiratory infection in juvenile offspring.

We also show that early gestational influenza affects hematopoiesis in offspring. Flow cytometric analysis revealed that frequencies of myeloid progenitor cells were increased in the bone marrow of offspring born to IAV-infected mice. Notably, these mice were fully naïve when sacrificed for analysis showing that the effects were due to in utero exposure. In addition, we observed abnormal frequencies of various functional immune cells in the offspring's lung and spleen. We speculate that reduced NK and B cell frequencies in the lungs from offspring born to IAV-infected dams might also contribute to increased vulnerability to a respiratory second hit but additional studies are required to examine these effects beyond the scope of the current study.

In contrast to chronic viral infections known to infect and replicate at immunological sites, acute infections such as influenza A virus infections are thought to perturb hematopoiesis indirectly through the action of cytokines as mediators (e.g. TNF)[35,36]. This appears to be the case with influenza-induced MIA in our model of early gestational influenza infection, being evident in both the lung and in plasma samples. In addition, we note the induction of TNF upon adoptive AM transfer that promoted viral clearance. Recent evidence suggests that blood monocytes can differentiate and migrate into distinct populations of human lung macrophages[37]. We speculate that altered hematopoiesis resulting from first hit may lead to impaired AM function and thus the ability to respond to postnatal second hits.

During fetal development, alveolar macrophages (AM) seed into the lung and remain long-lived therein mainly by self-renewal[26,38]. Although only subtle differences in AM frequencies were observed in the offspring of all groups, functional experiments in vivo and in vitro revealed that the functionality of alveolar macrophages is affected by maternal influenza. Improving the offspring's recovery after infection with IBV was possible by transferring AMs from control animals to offspring born to IAV-infected dams, prior to infection. These data suggest that alveolar macrophage function can be rescued by adding unaffected cells into the system. On the other hand, transfer of AMs from offspring born to MIA dams, did not affect disease outcome in control animals. It is important to highlight that we decided not to deplete host macrophages to avoid further disturbance of the AM niche and subsequent recruitment of new cells from the bone marrow into the lungs, prior to the experiment. Thus, we cannot exclude the possibility that some protective effects of healthy macrophages in offspring born to infected mothers might be due to an overall increase of alveolar macrophages in the lung. Nevertheless, our results provide strong evidence in favor of functional impairments in the macrophages of offspring caused by MIA, reflected by decreased viral clearance capacity in vitro and in vivo.

In summary, we provide evidence that early gestational influenza (first hit) increases offspring´s vulnerability towards other respiratory pathogens (second hits) particularly in early life. The underlying mechanisms seem to be multi-factorial. First, later life consequences of MIA correlate with increased IL-6, TNF, MCP-1, and IL-1ß levels in the lungs of IAV-infected dams. Second, increased early life vulnerability to second hits correlates with low birthweight. Third, alveolar macrophages of offspring exposed to maternal influenza show a reduced capacity to clear second hit pathogens. Our findings highlight the potential effects of infections in early pregnancy on the offspring to respiratory infections and emphasize the importance of measures that prevent maternal influenza.

## Methods

**Cells**. Cell lines of Madin-Darby Caine kidney cells II (MDCK II) were grown in modified Eagle's medium (MEM) supplemented with 10% fetal bovine serum (FBS), penicillin/streptomycin [0.1 mg/ml], and L-Glutamine [2 mM] and were cultivated at 5% $CO_2$, 96% rH, and 37 °C.

**Viruses**. The pH1N1 2009 influenza strain A/Hamburg/NY1580/09 was grown on MDCK II cells at 37 °C using a multiplicity of infection (MOI) of 0.2. Virus was harvested after 36 h. Virus titration was performed 72 h post infection on MDCK II cells by plaque assay. The influenza B virus B/Lee/40 was grown in embryonated SPF eggs at 33 °C for 3 days. In all, 100 µl of virus containing 200 PFU were injected into the allantoic fluid of eggs. The virus was harvested by aspirating the allantoic fluid after euthanizing the embryo at 4 °C for 4 h. All subsequent steps were performed on ice or at 4 °C. The allantoic fluid of individual eggs was tested for the presence of virus by hemagglutination assay and positive aliquots were pooled. The virus was purified

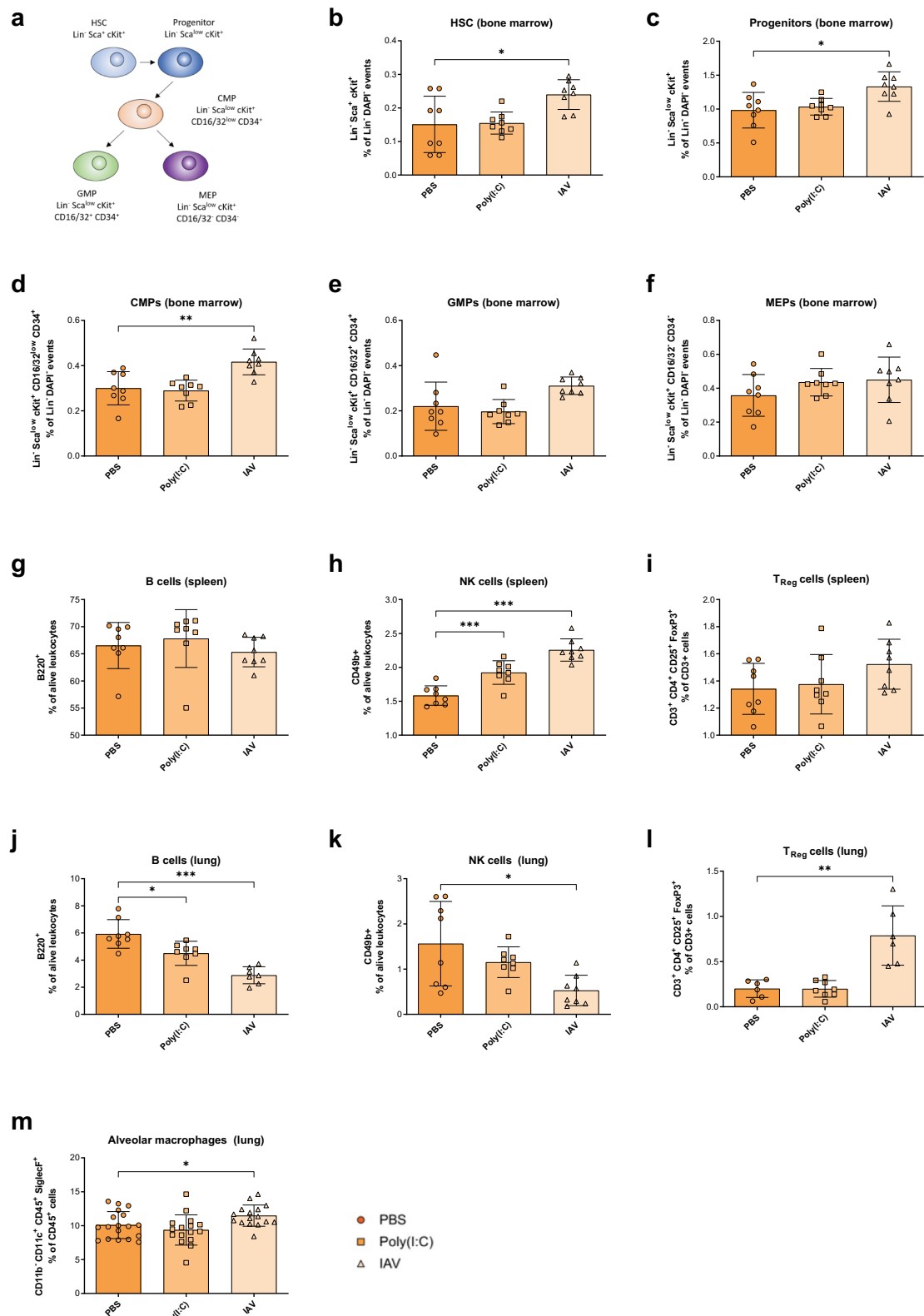

by centrifugation of the allantoic fluid at 5000 × g and 4 °C for 10 min and isolation of the supernatant. Virus aliquots were stored at −80 °C. All experiments of this study were performed using the same batch of the respective virus.

**Bacterial strains.** Methicillin-resistant *Staphylococcus aureus* (MRSA) were grown from a CryoBank stem conservation system JB1 for *S. aureus* strain USA300. A

stem plate was prepared by plating one unit from the stem conservation system onto a TS-agar plate and overnight incubation at 37 °C. Stem plates were kept for further use at 4 °C for up to 3 months. Pre cultures were prepared by transferring one colony from a stem plate into 2 ml of trypticase soy broth medium and incubation overnight at 37 °C while shaking at 180 rpm. Main cultures for infection were started by preparing a 1:100 dilution of the pre culture in TSB medium. The main culture was incubated at 37 °C and 180 rpm until the bacteria concentration

**Fig. 4 Altered hematopoiesis of offspring from influenza infected dams. a** Markers used to define respective stem (HSC) and progenitor cell populations by flow cytometry in this study. **b–f** Frequency of Lin⁻ Sca⁺ cKit⁺ HSC, $p = 0.0236$ (**b**); Lin⁻ Sca$^{low}$ cKit⁺ progenitor cells, $p = 0.0121$ (**c**); Lin⁻ Sca$^{low}$ cKit⁺ CD16/32$^{low}$ CD34⁺ common myeloid progenitor (CMP) cells, $p = 0.0035$ (**d**); Lin⁻ Sca$^{low}$ cKit⁺ CD16/32⁺ CD34⁺ granulocyte–monocyte progenitor (GMP) cells, $p = 0.0518$ (**e**); and Lin⁻ Sca$^{low}$ cKit⁺ CD16/32⁻ CD34⁻ megakaryocyte-erythrocyte progenitor (MEP) cells (**f**) as percentage of Lin⁻ DAPI⁻ events in the bone marrow of 2-week-old offspring ($n = 8$) born to early gestational phosphate-buffered saline (PBS)-treated, polyinosinic:polycytidylic acid (poly(I:C))-treated or influenza A virus (IAV)-infected dams, as assessed by flow cytometry. **g–l** Frequency of B220⁺ B cells (**g** and **j**, $p = 0.114$ poly(I:C), $p < 0.0001$), CD49b⁺ natural killer (NK) cells (**h**, $p = 0.0008$ poly(I:C), $p < 0.0001$ and **k**, $p = 0.0167$), CD3⁺ CD4⁺ CD25⁺ FoxP3⁺ regulatory T (T$_{Reg}$) cells (**l**, $p = 0.0710$ and **l**, $p = 0.0059$) as percentage of alive leukocytes in spleens (**g–i**) or lungs (**j–l**) of 2-week-old offspring ($n = 8$) born to early gestational PBS-treated, poly(I:C)-treated or IAV-infected dams, as assessed by flow cytometry. **m** Frequency of CD11b⁻ CD11⁺ CD45⁺ SiglecF⁺ alveolar macrophages as percentage of CD45⁺ cells in lung of 2-week-old offspring born to early gestational PBS-treated ($n = 18$), poly(I:C)-treated ($n = 16$) or IAV-infected ($n = 16$, $p = 0.0287$) dams, as assessed by flow cytometry. All $n$ represent number of male and/or female offspring from respective groups. Groups were merged for clarity as no difference between males and females was observed. All data are presented as mean and SD. Different groups of offspring are depicted in dark circles (PBS), medium squares (Poly(I:C)), or light triangles (IAV) in orange colors. The statistical significance was calculated by Welch's t test (two-tailed) (*$p < 0.05$, **$p < 0.01$, ***$p < 0.001$). PBS-treated groups were used as a reference and compared to IAV-infected groups in all statistical analyses unless stated otherwise. Source data are provided as a Source Data file.

reached $1.42 \times 10^8$ colony-forming units (CFU)/ml. Bacterial concentration was measured by optical density (OD600) that was found to correlate to the desired concentration by previous experiments. For infection, the bacteria were concentrated and subsequently diluted in PBS to a final dose of $1 \times 10^8$ CFU per animal. The final concentration was confirmed by re-titration.

**Mice.** In this study female C57BL/6J mice and male BALB/c mice were used for allogeneic breeding. All mice were bred in the in-house facility of the Leibniz Institute for Experimental Virology following standard protocols. All mice were kept in open cages (Type II long) at 22 °C (±0.3 °C) room temperature and a relative humidity of 45% (±10%) with a constant 12 h light–dark cycle. Food and water were available ad libitum all times. At the German Mouse Clinic mice were maintained in IVC cages (Type GM500) with water and standard mouse chow according to the directive 2010/63/EU, German laws and GMC housing conditions (www.mouseclinic.de). All mice that were not pregnant and older than 3 weeks received full maintenance food and pregnant mice as well as offspring until weaning received full breeding food. Specified pathogen-free (SPF) status of the facility and all animals was continuously confirmed every 3 months. All mice were kept for an acclimatization period of 14 days before start of breeding or experiments. All non-pregnant mice were kept in groups of 3–5, except for males that were used for breeding who were kept single-housed for the duration of breeding. In all, 8- to 10-week-old females were mated with 8- to 12-week-old males overnight and pregnancy was confirmed by plug-check after 16 h. Pregnant females were kept single-housed in open cages. Non-pregnant mice were continuously used for breeding. Females with a false-positive or false-negative pregnancy status were excluded from the experiments immediately. Offspring were kept with their mother until weaning and were separated by sex after weaning at 21 days post-partum (d p.p.). For offspring experiments, pups from at least 2 mothers were matched into the same experimental group for each condition. Additionally, all offspring experiments were performed and analyzed with respect to the offspring's sex.

All animal experiments were performed in compliance to relevant ethical considerations. Protocols were approved by the relevant German authorities (Behörde für Gesundheit und Verbraucherschutz Hamburg, approval number 124/12, 75/17, 97/19 and Behörde für Gesundheit, Verbraucherschutz und Pharmazie, Regierung von Oberbayern, approval number 46–16).

**Animal experiments.** In all, 8- to 10-week-old female C57BL/6J mice were mated with 8- to 12-week-old male BALB/c mice. Using standard protocols, pregnant or non-pregnant females were intranasally infected at gestational day 5.5 (E5.5) with $10^3$ PFU of the 2009 pH1N1 virus strain A/Hamburg/NY1580/09 or applied 4 mg/kg polyriboinosinic-polyribocytidilic acid (poly(I:C), Sigma) intraperitoneally or intranasally. Mice receiving Dulbecco's PBS (PBS, Sigma) intranasally were used as control group. Humane endpoints were defined according to the approved animal project license. In detail, animals were euthanized upon 25% reduction of relative weight gain compared to PBS-treated dams for pregnant animals and a 25% reduction of body weight compared to initial weight for non-pregnant animals. Some mice were monitored for weight loss and signs of disease until 14 d p.i. Additional groups of mice were euthanized on 1, 3, 6 d p.i. for organ harvest or at E17.5 (12 d p.i.) for assessment of pregnancy outcome and morphology of the fetuses. Pregnant mice that were not euthanized during pregnancy were kept single-housed in open cages and gave birth. The offspring was kept with the mother until weaning (21 d p.p.). For experiments with foster mothers, whole litters of neonates were exchanged between mothers with matching litter size (8 neonates) within 12 h after birth. Neonates were marked via tail tip painting and individually weighed every day until 14 days post-partum. For second-hit experiments, 2- and 6-week-old offspring of both sexes were infected with $10^5$ PFU of the influenza B virus (IBV) strain B/Lee/40 or $10^8$ CFU of methicillin-resistant *Staphylococcus aureus* (MRSA, USA 300), intranasally. Mice receiving PBS were used as control

group. Second-hit experiments for homologous and semi-homologous challenge were performed by applying $10^3$ PFU of the 2009 pH1N1 virus strain A/Hamburg/NY1580/09 or $10^2$ PFU of a H3N2 Aichi 6+2 reassortant (WSN backbone) intranasally. Mice receiving PBS were used as control group. Some mice infected with IBV were monitored for weight loss and signs of disease until 14 d p.i. Additional groups of these mice were euthanized on 1, 3, and 6 d p.i. for organ harvest. Mice that received MRSA were euthanized on 1, 3, and 6 d p.i. for organ harvest and were monitored for weight loss and survival until humane endpoints were reached. For flow cytometric analysis of immune cells, 2- and 6-week-old offspring born to treated and control mice were euthanized for organ explants without further treatment. For the adoptive transfer of alveolar macrophages, 2-week-old offspring born to prenatally infected or control treated mice were euthanized on day 14 d p.p. for isolation of alveolar macrophages (AMs) by FACS. Purified AMs ($10^5$ cells in 20 µl) were applied to 2-week-old mice intranasally. Offspring born to PBS-treated mothers received AMs either from offspring born to poly(I:C)- or IAV-treated mothers. Offspring born to poly(I:C)- or IAV-treated mice received AMs from PBS-treated mice. AMs were exchanged between offspring of the same sex. In total, 16 h after transfer of AMs, one cohort per group and sex was infected with $10^5$ PFU of IBV following standard protocols. Another cohort of each group and sex received PBS as control. All mice were monitored for signs of disease and were euthanized on day 3 p.i. for organ explants. Blood taken during animal experiments was centrifuged for 15 min at 3000×$g$ and 4 °C, and the respective plasma was stored at −80 °C. Organs were stored in 4% formalin solution at 4 °C or dry at −80 °C for subsequent homogenization. Sex of mice were determined by PCR (fetuses and neonates only) or by visual identification.

Assessment of morbidity was based on scoring criteria defined within the respective animal license approval as described in Supplementary Table 1.

**Lung function.** Mice were anesthetized with ketamine-xylazine, tracheostomized, and the lung function analyzed. In brief, respiratory function was measured using a FlexiVent system (Scireq). Mice were ventilated with a tidal volume of 10 ml/kg at a frequency of 150 breaths/min to reach a mean lung volume similar to that of spontaneous breathing. Testing of lung mechanical properties including dynamic lung compliance was carried out by a software-generated script that took four readings per animal.

**Virus titration.** Viral titers were measured by plaque assay. Six-well cell culture plates were seeded with 3 ml of MEM containing $2.0 \times 10^5$ MDCK II cells and incubated at 37 °C overnight. The virus was 10-fold serially diluted in PBS. After removing the media from the cells and washing them once with PBS, the cells were infected with 333 µl virus dilution or diluted organ homogenate per well and incubated at 33 °C (B/Lee/40) or 37 °C (pH1N1 2009) for 1 h. The plates were agitated every 10 min to ensure even distribution of the virus. After incubation, the virus supernatant was removed and overlay medium containing 1.25% Avicel and 0.1% TPCK-Trypsin in MEM was added to the plates. The cells were incubated 3 days at 37 °C or 4 days at 33 °C for pH1N1 2009 or B/Lee/40 respectively. After incubation, the cells were fixed in 4% paraformaldehyde solution at 4 °C for 30 min to 24 h. Plaques were visualized by either counterstaining with crystal violet solution (B/Lee/40) or by using IAV nucleoprotein-specific antibody staining and development via a horseradish conjugated secondary antibody and peroxidase substrate.

For IBV NP quantification via quantitative real-time PCR, vRNA was isolated from lung homogenates using the InnuPrep RNA MiniKit (Analytik Jena, Germany; catalog number: 845-KS-2040250) following the manufacturer's instructions. cDNA was generated using random nonamere primers (pd(N)9, Gene Link) and the SuperScript III kit (Thermo Fisher; catalog number: 18080051). qRT-PCR was performed at 95 °C (600 s), 95 °C (15 s), 55 °C (10 s), and 72 °C (20 s) with 45 cycles. For IBV NP primers used, see key resource table. cT values were

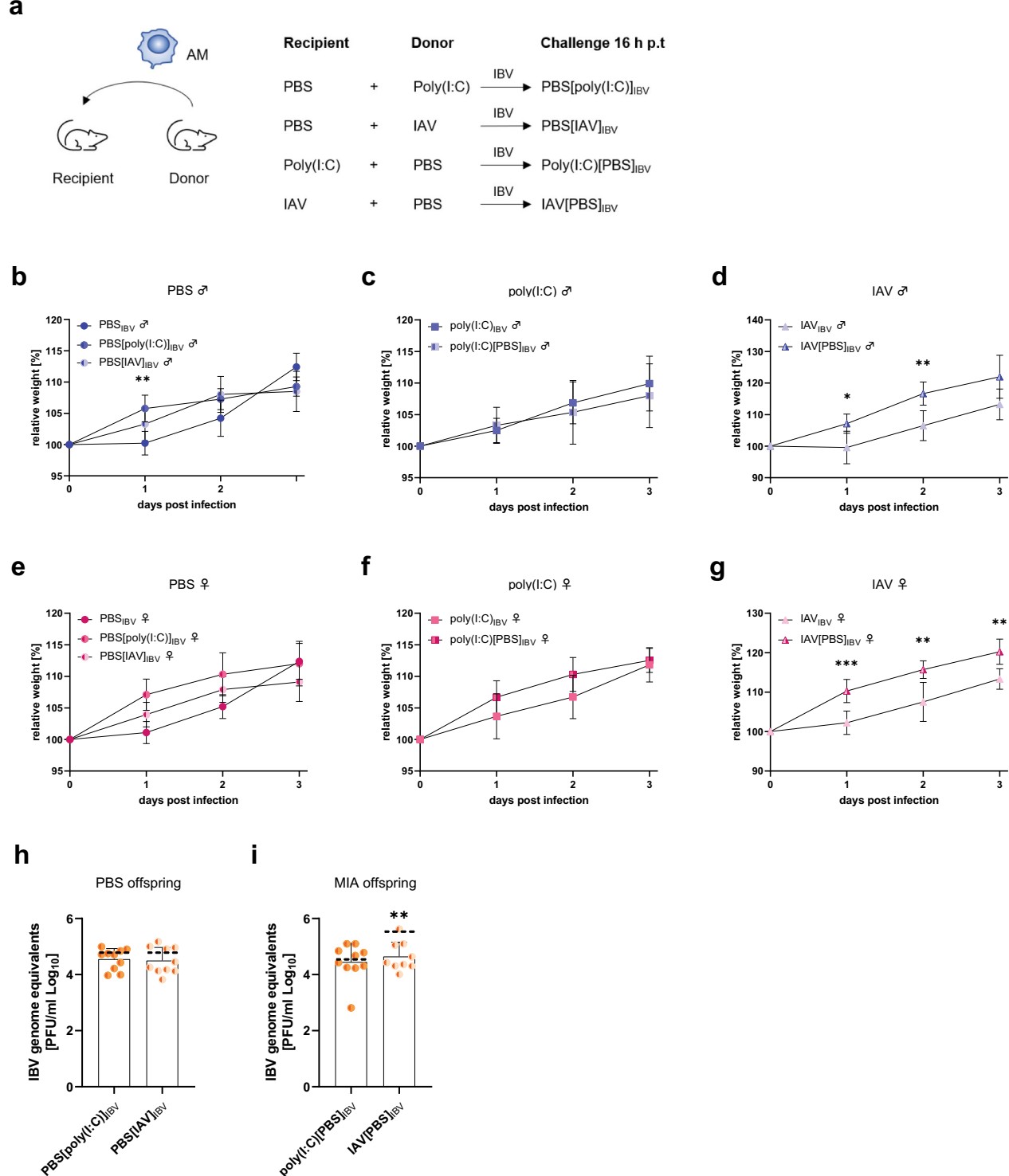

correlated to cT values of input IBV titers defined by plaque test. Unites were defined as genome equivalents. Primers used are shown in Supplementary Table 2.

**Bacterial titration.** The bacterial load in mouse lung tissue was measured by plating tissue homogenates onto mannitol salt phenol red (MS) agar plates. Organ homogenates were prepared fresh by homogenizing the lung tissue in 1 ml of PBS in a bead mill (Mixer mill MM400,Retsch) at 30 Hz and 4 °C for 10 min. In all, 10-fold serial dilutions of the homogenates were prepared in ice-cold PBS. The dilutions were plated onto 10 cm Petri dishes in hot MS agar (65 °C). All measurements were performed in technical duplicates. MS agar plates were incubated at 37 °C for 2 days before MRSA colonies were identified by shape and color change of the agar.

**Immunohistology and histopathology.** Lungs of infected animals were processed for histopathological examination and immunohistochemistry according to established protocols. Deparaffinized tissue sections were treated with 0.1 M citrate buffer (pH 6.0) and a rabbit anti-H7N1 serum. The primary antibody (anti-IAV NP) was detected by a biotin-conjugated anti-rabbit antibody (Jackson Immu-noResearch) followed by the application of the Zytochem-Plus HPR kit (Zytomed; catalog number: ZUC031-006). Tissues were counterstained with hematoxylin for pathological analysis.

**Flow cytometry and fluorescence activated cell sorting.** Lung tissue was minced on ice and incubated in lysis buffer (2.5 mg/ml Collagenase D, 0.01 mg/ml DNase I in RPMI) at 37 °C for 30 min with constant agitation. Bone marrow was isolated by centrifugation from mouse tibia and femur at $1000 \times g$ for 5 min at 4 °C. Single cell

**Fig. 5 Functionally impaired alveolar macrophages in offspring of influenza infected dams. a** Adoptive transfer experiments were performed in 2-week-old offspring followed by influenza B virus (IBV) infection 16 h post transfer (h p.t.). **b, e** Alveolar macrophages (AM) of 2-week-old male, $p = 0.0058$ (**b**) and female (**e**) offspring born to polyinosinic:polycytidylic acid (poly(I:C))-treated mothers ($n = 10$ males, 10 females) or influenza A virus (IAV)-infected mothers ($n = 11$ males, 14 females) were pooled, and $10^5$ AMs were transferred intranasally into offspring of the same sex ($n = 5$ [male, poly(IC)], 6 [male, IAV], 8 [female, poly(I:C)], 7 [female, IAV]) born to phosphate-buffered saline (PBS)-treated mothers with subsequent infection of the sentinel-offspring 16 h post transfer with $10^5$ plaque forming units (PFU) of IBV. **c, d, f, g** Alveolar macrophages of 2-week-old male (**c, d**) and female (**f, g**) offspring born to PBS-treated mothers ($n = 22$ males, 29 females) were pooled and $10^5$ AMs were transferred intranasally into offspring of the same sex ($n = 7$ [male, poly(I:C)], 8 [female, poly(I:C)], 6 [male IAV, $p = 0.0115$ (1 dp.i.), $p = 0.0050$ (2d p.i.)], 6 [female IAV, $p = 0.008$ (1d p.i.), $p = 0.0068$ (2d p.i.), $p = 0.0071$ (3d p.i.)]) born to poly(I:C)-treated (**c, f**) or IAV-infected (**d, g**) mothers with subsequent infection of the sentinel-offspring 16 h post transfer with $10^5$ PFU of IBV. Weight development was observed until 3 d p.i. **h** IBV lung titer from offspring born to PBS-treated dams and after transfer of AMs from poly(I:C)- or IAV-offspring with subsequent IBV challenge 12 h p.t. at 3 d p.i. ($n = 10$). Dotted lines indicate reference titers from IBV infected offspring without AM transfer (shown in Fig. 3). **i** IBV lung titer from offspring born to poly(I:C)-treated ($n = 10$) or IAV-infected, $p = 0.089$ ($n = 9$) dams and after transfer of AMs from PBS-offspring with subsequent IBV challenge 12 h p.t. at 3 d p.i. All experiments were also performed with PBS-treatment of the offspring post AM transfer but data are not shown to enhance clarity and all PBS-treated offspring were negative for reduced weight gain or virus titers, as expected. All data are presented as mean and SD. Different groups of male (♂) offspring are depicted in dark circles (PBS, medium squares (Poly(I:C)) or light triangles (IAV) in blue colors and female (♀) offspring in red colors. Different groups of offspring that were not stratified by sex are depicted in orange colors with the same icons. The statistical significance was calculated by two-tailed $t$-tests using the Bonferroni-Dunn multiple testing correction (**b–g**) or by Welch's $t$ test (two-tailed) (**h, i**) (*$p < 0.05$, **$p < 0.01$, ***$p < 0.001$). Non-significant comparisons are not depicted within the respective figures. PBS-treated groups were used as a reference and compared to IAV-infected groups in all statistical analyses unless stated otherwise. Source data are provided as a Source Data file.

solutions were prepared using a 70-μm cell strainer (Corning) in FACS buffer (2% FCS in PBS). Red blood cells were lysed using RBC lysis buffer (BioLegend) for 5 min on ice. Life/dead staining was performed using Zombie Nir Fixable Viability Kit (BioLegend; catalog number: 423105) for alveolar macrophages or DAPI for other populations following the manufacturer's instructions. Antibody staining was performed for 30 min at RT in the dark with an antibody concentration of 6 μl/ml. For antibody details see key resource table. Stained cells were resuspended in FACS buffer containing 2% BSA for subsequent analysis or sorts. Flow cytometric analysis as well as FACS was performed using a FACSAria II Fusion SORP (BD), analysis was performed on a FACSCantoII and data analysis was done using FACSDiva software v 8.0 as well as FCS Express 6 (DeNovo Software, USA). A representative example gating strategy that was used for sorting AMs and the confirmed purity of the sort (re-analysis) is shown in Supplementary Fig. 9. Representative gating strategies used for the data shown in Fig. 4 are shown in Supplementary Fig. 10.

**Cytokine and hormone measurement**. Cytokines were measured in supernatants of homogenized lungs or plasma, collected from mice at 1, 3, or 6 d p.i. Therefore, lung tissue (~100 mg) was homogenized in 1 ml of PBS in a bead mill (Mixer mill MM400, Retsch) at 30 hz for 10 min at 4 °C. The cytokines (MCP-1, IL-1β, IL-2, IL-6, IL-10, IL-17A, and TNF) were then detected using a custom-made Bio-Plex Pro Mouse Cytokine 17-plex assay (Bio-Rad; catalog number: #M5000031YV) following the manufacturer's instructions in a Bio-Plex 200 System with high-throughput fluidics (HTF; Bio-Rad). All measurements were run in duplicate. Progesterone (Cayman Chemical; catalog number: 582601) and corticosterone (ARBOR ASSAYS; catalog number: K014-H1) levels in maternal serum were evaluated by ELISA following the manufacturer's instructions. For progesterone analysis, serum was diluted 1:100 or 1:200 and measured after 70 min of substrate incubation. All ELISAs were measured on a Saphire2 ELISA microplate reader (Tecan) and evaluated using a four-parameter logistic regression (MyAssays). To measure serum testosterone levels, a chemiluminescence immunoassay (ADVIA Centaur Testosterone II assay; Siemens Healthcare Diagnostics; catalog number: 10696862) was employed. Measurement was performed with the ADVIA Centaur XP (Siemens Healthcare Diagnostics).

**Antibodies and reagents**. All antibodies and reagents used in this study are listed in Supplementary Table 2.

**Nutrient gene RT-qPCR**. Nutrient gene expression was measured from RNA of homogenized placental tissue taken at E17.5. RNA was isolated using the Analytik Jena kit Innuprep 2.0 following the manufacturer's instructions with additional on-column DNase I-treatment (Roche). cDNA was generated using the Superscript III Reverse Transcriptase kit (Invitrogen) using random nonamere primers d(N)9 (Gene Link). RT-qPCR was performed to analyze placental gene expression levels of two labyrinthine-specific, growth stimulatory genes, the growth inhibitory *Grb10* (growth factor receptor bound protein 10) and the growth stimulatory gene *Igf2* (insulin-like growth factor II) and two isoforms of the system A family of amino acid transporters, *Slc38a1* (solute carrier family 38, member 1) and *Slc38a2* (solute carrier family 38, member 2), which are responsible for transplacental transport of neutral amino acids[31]. DNA oligonucleotides for amplification of the target genes as well as the housekeeping gene *Ywhaz* (Tyrosine 3-Monooxygenase/ tryptophan 5-monooxygenase activation protein, zeta polypeptide) were designed using Primer-

BLAST (http://www.ncbi.nlm.nih.gov/tools/primer-blast/index.cgi). Singleplex reactions (20 μl) were set up in $H_2O$ PCR grade (Roche) and FastStart Essential DNA Green Master (2x, Roche) with 300 nM of forward and reverse primer each, and 2 μl cDNA template. RT-qPCR runs were conducted on the LightCycler 96 with end-point fluorescence detection for SYBR Green: 10 min at 95 °C, 50 amplification cycles (15 s at 95 °C, 10 s at 65 °C, and 20 s at 72 °C). Subsequently, melting curve analysis was performed (15 s at 95°C, 15 s at 60 °C, and 1 s at 95 °C). Analysis was performed in triplicate for each GOI in each sample. Data were subsequently analyzed using LinReg PCR software and GOI expression for PBS sample of each gene were set as 1. The relative N0(GOI)/N0 (*Ywhaz*)-expression values of the biological replicates are presented. Primers used are shown in Supplementary Table 2.

**Sex determination by PCR**. The sex of mouse fetuses was determined by PCR using SX primers[39]. Primers used are shown in Supplementary Table 2.

**Quantification and statistical analysis**. All data were analyzed using Graph Pad Prism v.8.4.2. Statistical details of all experiments are indicated in the respective figure legends. Level of statistical significance was defined as $p < 0.05$ (*$p < 0.05$; **$p < 0.01$; ***$p < 0.001$). Data points that were defined as outliers by Grubb's test were excluded from further analysis. Datapoints that were designated technical failures (e.g., no reads by the machine or if pipetting problems occured) were removed. Groups of biological replicates in this study were defined as being sampled from a Gaussian distribution by biological consideration of the respective experiments. No tests were performed to check for Gaussian distribution.

**Reporting summary**. Further information on research design is available in the Nature Research Reporting Summary linked to this article.

## Data availability

All relevant data are available within the paper or its Supplementary Information files or from the corresponding author upon reasonable request. Source data are provided with this paper.

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

## Acknowledgements
We thank Oliver Strauch and Ursula Müller from the technology platform small animal models at the Leibniz Institute for Experimental Virology, for their excellent support. This study was funded by the Deutsche Forschungsgemeinschaft (DFG.; GA 1575/3-2 to G.G.), the German Free and Hanseatic City of Hamburg (to G.G.), and the German Ministry of Health (to G.G.). S.L. is funded by the European Research Council (ERC) under the European Union's Horizon 2020 research and innovation program (Grant Agreement No. 758713) and by the Hector Stiftung II. P.O. receives support from an NIHR Senior Investigator Award NIHR201385 (Award: WHRR P84026); the NIHR Imperial Biomedical Research Centre (BRC) IS-BRC-1215-20013 and EU FP7 PREPARE project 602525.

## Author contributions
G.G. and H.J. designed the study and all experiments. H.J., K.W.G., N.T.B., L.S., V.P.R., A.G., and H.J. performed animal experiments. I.B.B., N.B., J.B., A.D., S.L., and B.S. performed and analyzed FACS assays. G.P.S. performed immunohistochemical analysis. N.M.K. performed PCR assays. A.J., A.Ö.Y., H.F., V.G.D., C.S., M.H.A., N.M.K., M.Z., M.W., and S.K.E. assessed lung functions and analyzed data. K.K. and T.M. performed histopathological analyses. F.C. and P.O. designed experiments with alveolar macrophages. A.K. performed statistical analysis. H.J. and G.G. analyzed data and wrote the manuscript. All authors revised the manuscript.

## Funding

## Competing interests
The authors declare no competing interests.

## Additional information

[1]Department of Viral Zoonoses – One Health, Leibniz Institute for Experimental Virology, Hamburg, Germany. [2]Department of Oncology, Hematology and Bone Marrow Transplantation with Section Pneumology, Hubertus Wald Comprehensive Cancer Center Hamburg, University Medical Center Hamburg-Eppendorf, Hamburg, Germany. [3]Department of Tumor Biology, Center of Experimental Medicine, University Medical Center Hamburg-Eppendorf, Hamburg, Germany. [4]Early Life Origins of Chronic Lung Disease, Research Center Borstel, Leibniz Lung Center, Member of the German Center for Lung Research (DZL), Borstel, Germany. [5]Department of Computational Biology of Infection Research, Helmholtz Centre for Infection Research, Braunschweig, Germany. [6]Flow Cytometry/FACS Unit, Leibniz-Institute for Experimental Virology, Hamburg, Germany. [7]Microscopy and Image Analysis Unit, Leibniz-Institute for Experimental Virology, Hamburg, Germany. [8]Comprehensive Pneumology Center (CPC), Institute of Lung Biology and Disease, Helmholtz Center Munich, German Research Center for Environmental Health, Neuherberg, Germany. [9]Member of The German Center for Lung Research (DZL), Borstel, Germany. [10]German Mouse Clinic, Institute of Experimental Genetics, Helmholtz Center Munich, German Research Center for Environmental Health, Neuherberg, Germany. [11]Chair of Experimental Genetics, School of Life Science Weihenstephan, Technische Universität München, Freising, Germany. [12]German Center for Diabetes Research (DZD), Neuherberg, Germany. [13]Institute for Pathology and Neuropathology, University Hospital Tübingen, Tübingen, Germany. [14]National Heart and Lung Institute, Imperial College London, London, UK. [15]Core Facility Fluorescence Cytometry, Research Center Borstel, Leibniz Lung Center, Borstel, Germany. [16]Division of Personalized Medical Oncology (A420), German Cancer Research Center (DKFZ), Heidelberg, Germany. [17]Department of Personalized Oncology, University Hospital Mannheim, University of Heidelberg, Mannheim, Germany. [18]Junior Research Group Coinfection, Priority Research Area Infections, Research Center Borstel, Leibniz Lung Center, Borstel, Germany. [19]Institute of Experimental Medicine, Christian-Albrechts-Universität zu Kiel, Kiel, Germany. [20]Institute for Virology, University for Veterinary Medicine Hannover, Hannover, Germany. [21]German Center for Infection Research (DZIF), Braunschweig, Germany. ✉email: guelsah.gabriel@leibniz-hpi.de

