## [Peer Review File · Nature Communications]

REVIEWER COMMENTS

Reviewer #1 (Remarks to the Author):

This is a report by Jacobsen and colleagues about the impact of early gestation influenza virus infection on subsequent infection with a heterosubtypic influenza strain or Methicillin-resistant *S. aureus*. Strengths of the study are the use of a novel in vivo animal model to recapitulate first trimester infection, investigation of sex-specific differences in the survival and immune response to fetal infection and use of both a viral and bacterial secondary infection. Overall, the studies are well-planned out with sufficient experimental replicates and power for appropriate statistical analysis. The primary finding is that fetuses exposed to maternal influenza virus infection are more susceptible to both a second flu or MRSA infection after birth. The premise is convincing because this is certainly documented in human neonates. Neonates exposed to maternal infection in utero have worse outcomes after delivery.

1. Line 131: There needs to be an explanation for why the BALB/c-mated allogenic pregnant C57BL/6 female scheme was used. Of note, it is C57BL/6, not C57Bl/6.

2. Lines 140-144: The data in Figure 1 does not support the statements made here, specifically the impact of poly I:C on plasma cytokine levels. The only statistically significant impact is on MCP-1 at 1-day post infection. In addition, the axis scale on the figures is not consistent, which makes it difficult to compare the different cytokines.

3. Figure 2 D is missing labels under the treatment groups, like 2C. The legend indicates that it should have the same labels as 2C. Also, on line 180, you state that there was "only a slight increase in male offspring was observed for dams that were treated with poly I:C". The first set of bars has the poly I:C group 60% female and 40% male. The second poly I:C group is 50% male. Therefore, I am not sure how you are drawing this conclusion.

4. Labels are also missing in 2G, like 2D

5. Lines 195-196: This is a strong statement considering the pulmonary function tests were done at 20 weeks of age. Considering the flu exposed fetuses remained growth restricted at 6 weeks of age (Figure 2i), it would be of greater clinical significance to do the PFTs at this age, particularly since the second hit is done much earlier than 20 weeks of age.

6. Lines 244-245: There is a statement that the animals died before sex determination was possible, which seems confusing because sex determination was possible with the flu infected 2-week old animals. Was there cannibalization of only the bacteria infected animals?

7. Lines 247-248: Again, there is an overstatement of the results in Figure 3 e-j. There is only one comparison which is statistically significant, IL2 at 3 DPI so it seems like it is a stretch to say the increased vulnerability of offspring... with increased inflammatory cytokine response early in infection".

8. It would be helpful to have labels above the three survival graphs in Figure 3b-d.

9. There should be representative flow plots and the gating strategy for Figure 4.

10. Line 339: I think it is important to state that it is an intranasal adoptive transfer, as this is not the traditional route (either IV or IP). How was the 10^5 dose determined?

11. Again, labels on 5b-g would be very helpful. Why is the weight gain only reported for the first three days after infection?

12. Style and grammar: Line 193, 335: There is improper use of a comma: "...treated mothers immediately after birth revealed that this effect is not..." This happens in other places throughout the manuscript and is distracting to an otherwise well-written manuscript. Also, please check the word pup throughout the manuscript. There are a couple of times where it is pub (line 256, 560). Line 308: do you mean on the other hand?

Sincerely,
Alison Carey, MD
Drexel University College of Medicine

Reviewer #2 (Remarks to the Author):

In this manuscript, Jacobsen et al found that offspring born to influenza A virus (IAV) infected mothers were more vulnerable to secondary heterologous influenza B virus (IBV) and *Staph aureus* (MRSA) infection. Interestingly, they showed that poly IC treatment, which only transiently

engaged innate antiviral immune responses, did not have the same effects as those of IAV infected moths. They further showed that early gestational IAV infection likely affected offspring hematopoiesis and alveolar macrophage (AM) function, which might underlie the enhanced susceptibility to IBV and/or MRSA infection in the offspring. While some of the phenotypic findings in Fig. 1-4 are of interest, there are a couple of key issues in the manuscript and need to be addressed.

First, the experiments in figure 5 were rather poorly performed. The authors showed that IAV infection in early gestation age led to the alteration of monocyte development at two weeks of age. However, it is pretty clear now that AM development is independent of hematopoietic system after birth. Therefore, the rationale to examine AM function in Fig. 5 is not justified. Further, the authors claimed that the functionality of AMs was impaired in the offspring of IAV-infected mothers, but was not supported by their data. In the experiments of AM transfer, they did not provide proper controls to directly compare the effects of transfer of AMs from offspring of IAV-infected mothers to those AMs from offspring of PBS/Poly IC-treated mothers (The results observed could be simply due to the presence of extra numbers of AMs as AMs are protective during IAV infection). The use of the *in vitro* culture experimental system is not scientifically supported and only three mice were used.

Second, the authors indicated that the enhanced susceptibility to IAV or MRSA infection was likely due to increased inflammation. However, the differences in the three cytokines shown among the experimental groups were quite moderate. The authors should perform a multiplex analysis to examine more inflammatory cytokines. Furthermore, a kinetic virus burden measurement should be provided to support their argument.

Third, the result section lacks sufficient experimental details. It is pretty hard to judge the rationale of their experimental design and the soundness of the data.

Reviewer #3 (Remarks to the Author):

In this paper, the authors show that offspring born to influenza infected pregnant mice display increased susceptibility to viral and bacterial respiratory infection. This effect associates with alteration in the frequencies of hematopoietic stem cells and with altered functions of alveolar macrophages. The transfer of alveolar macrophages collected from naive offsprings to offsprings born to influenza infected pregnant mice protects the later against lethal influenza (B) virus infection.

Globally the paper is very interesting and adds novelty in the field. On the other hand, it is too immature at this stage. More mechanistic insights should be presented. There are also others issues.

In Figure. 1, IAV-infected non pregnant mice should be shown as controls. Panel A is not clear (who is who?). In panel D, I am surprised that the viral load remains constant at day 6 p.i. The lack of inflammatory cytokines in p(I :C)-inoculated mice is not surprising. What happens after i.n. (and not after i.p) p(I :C) inoculation, which appears to be a better control? The increased concentrations of corticosterone and progesterone in dams infected with IAV is not convincing. Have the authors analyzed the pathology as well (panel m is not commented).

Differences in Fig. 2E and 2F (at least the IAV group) are not convincing. Panel G, I wonder whether stats can be done between the p(I :C) group and the IAV group.

Sup Fig.2 mentions protective antibodies but this is not shown. Interesting to quantify. Data shown in Fig. 3E-J are not convincing. A mock control is lacking. In Fig. 3K (and not Sup 3K), there is a higher bacterial load (6 weeks) but this does not translate into higher pathogenicity. Please explain. Strange way to represent the bacterial loads. What happens in 2 weeks old offspring (bacterial/viral loads affected too ?). In general, do the authors think that after 6 weeks the enhanced susceptibility of offsprings is maintained ? Is it really a long-lasting response (line

423) ?

In Figure. 4, the number of animals should be increased to enhance statistics. Other progenitors should be investigated. Is there a difference in term of cell number ?

The experiment proposed in Fig . 5 is elegant and should be extremely difficult to accomplish. I however suggest to analyze other criteria (not only the weight). Data from the in vitro experiment are not extremely convincing in terms of differences. It would be interesting to perform RNAseq analysis of alveolar macrophages.

Many links are lacking in this ms. In general, the link between what happens in the BM and what happens in the lungs is lacking. The link between MIA and altered response of offsprings is also not shown. Do the authors think that alteration of the gut microbiota in infected pregnant mice participate in altered response of offsprings (altered microbiota has been shown to impact on hematopoiesis and on lung defence). FTE experiments are realisable. To conclude, more insight should be placed to enhance the quality of this very interesting story.

Other comments

The authors should mention the literature describing the impact of influenza on hematopoietic stem cells. In general, the text of the Figure legends is by far too long.

Mechanistically, the authors suggest alteration

Point-by-Point Response

Reviewer #1

*This is a report by Jacobsen and colleagues about the impact of early gestation influenza virus infection on subsequent infection with a heterosubtypic influenza strain or Methicillin-resistant *s. aureus*. Strengths of the study are the use of a novel in vivo animal model to recapitulate first trimester infection, investigation of sex-specific differences in the survival and immune response to fetal infection and use of both a viral and bacterial secondary infection. Overall, the studies are well-planned out with sufficient experimental replicates and power for appropriate statistical analysis. The primary finding is that fetuses exposed to maternal influenza virus infection are more susceptible to both a second flu or MRSA infection after birth. The premise is convincing because this is certainly documented in human neonates. Neonates exposed to maternal infection in utero have worse outcomes after delivery.*

We thank Dr. Carey for her comments and valuable criticism, which we have now all addressed as described below.

Specific points:

#1: 1. Line 131: There needs to be an explanation for why the BALB/c-mated allogenic pregnant C57BL/6 female scheme was used. Of note, it is C57BL/6, not C57Bl/6.

This information is included in the introduction and is highlighted again in the respective results section. C57Bl/6 was corrected to C57BL/6 throughout the entire manuscript.

#2. Lines 140-144: The data in Figure 1 does not support the statements made here, specifically the impact of poly I:C on plasma cytokine levels. The only statistically significant impact is on MCP-1 at 1-day post infection. In addition, the axis scale on the figures is not consistent, which makes it difficult to compare the different cytokines.

We have amended the respective section, now describing in detail the poly(I:C) induced cytokines compared to IAV induced cytokines. Regarding the axis, there are large differences between the units measured (100 versus 1000 units). Adjusted levels led to masking differences and therefore we decided to keep the initial format.

#3. Figure 2 D is missing labels under the treatment groups, like 2C. The legend indicates that it should have the same labels as 2C. Also, on line 180, you state that there was “only a slight increase in male offspring was observed for dams that were treated with poly I:C”. The first set of bars has the poly I:C group 60% female and 40% male. The second poly I:C group is 50% male. Therefore, I am not sure how you are drawing this conclusion.

We have amended the figure labels in the graph for Figure 2c and d. The reviewer is correct regarding the misleading statement that is now amended accordingly.

#4. Labels are also missing in 2G, like 2D

We have amended the figure label in the graph for Figure 2g as well.

#5. Lines 195-196: *This is a strong statement considering the pulmonary function tests were done at 20 weeks of age. Considering the flu exposed fetuses remained growth restricted at 6 weeks of age (Figure 2i), it would be of greater clinical significance to do the PFTs at this age, particularly since the second hit is done much earlier than 20 weeks of age.*

We thank the referee for this criticism and agree that such studies would significantly enhance our work. We have now repeated the requested lung function experiments by generating new cohorts of **6-week old offspring** born to IAV, poly(I:C) or PBS treated dams. The new data are now shown in the **new Supplementary Figure 2 a-d**.

#6. Lines 244-245: *There is a statement that the animals died before sex determination was possible, which seems confusing because sex determination was possible with the flu infected 2-week old animals. Was there cannibalization of only the bacteria infected animals?*

In this cohort, animals were found dead because of cannibalization (all animals were found dead and were not euthanized because of humane endpoints). Hence, carcasses were discarded without taking tissue samples for sex-determination. After this experiment, we started determining sex prior to infection for all subsequent experiments which allowed us to assess sex-differences in the IBV experiments.

#7. Lines 247-248: *Again, there is an overstatement of the results in Figure 3 e-j. There is only one comparison which is statistically significant, IL2 at 3 DPI so it seems like it is a stretch to say the increased vulnerability of offspring... with increased inflammatory cytokine response early in infection”.*

We have amended the respective statement accordingly.

#8. *It would be helpful to have labels above the three survival graphs in Figure 3b-d.*

We have included labels above Figure 3 b, c and d as requested.

#9. *There should be representative flow plots and the gating strategy for Figure 4.*

We now provide a **new Supplementary Figure 9** showing representative flow plots and gating strategies for the data shown in Figure 4.

#10. Line 339: *I think it is important to state that it is an intranasal adoptive transfer, as this is not the traditional route (either IV or IP). How was the 10^5 dose determined?*

We have amended the respective sentence accordingly. Flow cytometry experiments in 2-week-old offspring revealed, that we were repeatedly able to recover 10^5 AMs per animal. For

the transfer experiments, we wanted to use as many cells as possible to reveal possible rescue effects.

#11. Again, labels on 5b-g would be very helpful. Why is the weight gain only reported for the first three days after infection?

We have included the labels on figure 5-g as requested. We have concentrated on day 3 p.i. where the highest virus titer was detected. Since these experiments alone required n=300 animals with respective control groups, we did not assess later time points that would not have substantially increased knowledge.

#12. Style and grammar: Line 193, 335: There is improper use of a comma: "...treated mothers immediately after birth revealed that this effect is not..." This happens in other places throughout the manuscript and is distracting to an otherwise well-written manuscript. Also, please check the word pup throughout the manuscript. There are a couple of times where it is pub (line 256, 560). Line 308: do you mean on the other hand?

We have now carefully revised the manuscript to address the stylistic concerns.

Reviewer #2

In this manuscript, Jacobsen et al found that offspring born to influenza A virus (IAV) infected mothers were more vulnerable to secondary heterologous influenza B virus (IBV) and Staph aureus (MRSA) infection. Interestingly, they showed that poly IC treatment, which only transiently engaged innate antiviral immune responses, did not have the same effects as those of IAV infected moths. They further showed that early gestational IAV infection likely affected offspring hematopoiesis and alveolar macrophage (AM) function, which might underlie the enhanced susceptibility to IBV and/or MRSA infection in the offspring. While some of the phenotypic findings in Fig. 1-4 are of interest, there are a couple of key issues in the manuscript and need to be addressed.

We thank the reviewer for his/her comments and valuable criticism, which we have now all addressed as described below.

Specific points:

#First**, the experiments in figure 5 were rather poorly performed. The authors showed that IAV infection in early gestation age led to the alteration of monocyte development at two weeks of age. However, it is pretty clear now that AM development is independent of hematopoietic system after birth. Therefore, **the rationale to examine AM function in Fig. 5 is not justified**. Further, the authors claimed that the functionality of AMs was impaired in the offspring of IAV-infected mothers, but was not supported by their data. In the experiments of AM transfer, they did not provide proper controls to direct compare the effects of transfer of AMs from off-spring IAV-infected mothers to those AMs from offspring of PBS/Poly IC-treated mothers (**The results observed could be simply due to the presence of extra numbers of AMs as AMs are protective during IAV infection**). **The use of the in vitro culture experimental system is not scientifically supported and only three mice were used.

We have now performed additional experiments to strengthen the data shown in Figure 5. We determined virus titres after transfer of alveolar macrophages and viral challenge. The new data is now presented as **new Figure 5 h and i**. Additionally, we measured cytokine responses after adoptive transfer of alveolar macrophages before and after second hit. The new data are now presented in the **new Supplementary Figure 7**.

Regarding AM transfer: we do not agree with the criticism that simply extra AM numbers are showing protective effects against influenza. In Figure 5, we show that AM transfer (healthy AMs) into poly(I:C)-offspring and transfer of AMs (impaired AMs) into PBS-offspring does not alter disease outcome measured by weight loss (Figure 5 b-g). This finding is now further supported by the fact that IBV titers (new Figure 5h,i) in the respective control groups are unaffected by AM-transfer. Rescue of adverse effects is only observed after transfer of healthy AMs into IAV-offspring.

Regarding Figure 5j: in vitro co-infection experiments with AMs were performed as an additional method to assess the in vivo findings with respect to virus clearance. We agree that an n=3 represents the lower limit but therefore, we included both sexes revealing 2x n=3.

#Second**, the authors indicated that the enhanced susceptibility to IAV or MRSA infection was likely due to increased inflammation. **However, the differences in the three cytokines shown

among the experimental groups were quite moderate. The authors should perform a multiplex analysis to examine more inflammatory cytokines. Furthermore, a kinetic virus burden measurement should be provided to support their argument.

Thank you, we agree. We have now performed a 17-plex cytokine analysis and those showing detectable differences are now presented in the **new Supplementary Figure 1**. Yes, we also measured the virus burden as requested. The data are shown in the **new Figure 1e**.

#Third, the result section lacks sufficient experimental details. It is pretty hard to judge the rationale of their experimental design and the soundness of the data.

We hope that the Referee agrees that the new data provided and the revisions to the manuscript address this issue in full.

Reviewer #3

In this paper, the authors show that offspring born to influenza infected pregnant mice display increased susceptibility to viral and bacterial respiratory infection. This effect associates with alteration in the frequencies of hematopoietic stem cells and with altered functions of alveolar macrophages. The transfer of alveolar macrophages collected from naive offsprings to offsprings born to influenza infected pregnant mice protects the later against lethal influenza (B) virus infection.

Globally the paper is very interesting and adds novelty in the field. On the other hand, it is too immature at this stage. More mechanistic insights should be presented. There are also others issues.

We thank the reviewer for his/her comments and valuable criticism, which we have now all addressed as described below.

Specific points:

*#1: In Figure. 1, IAV-infected non pregnant mice should be shown as controls. Panel A is not clear (who is who?). In panel D, I am surprised that the viral load remains constant at day 6 p.i. The lack of inflammatory cytokines in p(I:C)-inoculated mice is not surprising. **What happens after i.n. (and not after i.p) p(I:C) inoculation, which appears to be a better control?** The increased concentrations of corticosterone and progesterone in dams infected with IAV is not convincing. Have the authors analyzed the pathology as well (panel m is not commented).*

We now include the requested data in non-pregnant mice (Figure 1 c). The viral titers shown in Figure 1d increase by almost 1log at day 3 p.i. and then decrease again 6 days p.i..

In addition, we have now performed additional requested experiments and have treated pregnant mice **intranasally** with poly(I:C) and measured the entire panel of cytokines in plasma and lung as shown previously for intraperitoneally poly(I:C) treated dams in Figure 1. The new data are now presented in the **new Supplementary Figure 1 a-o**.

We now find that intranasal poly(I:C) treatment is generally less potent at inducing cytokine responses compared to intraperitoneal poly(I:C) treatment. Therefore, we kept the i.p. treatment group as a better control in the main Figure 1.

We agree that the hormone levels are marginal. We have adjusted the respective sections in the results section accordingly.

Panel m is now described as panel d in Figure 1.

#2: Differences in Fig. 2E and 2F (at least the IAV group) are not convincing. Panel G, I wonder whether stats can be done between the p(I:C) group and the IAV group.

Data shown in Figure 2e are obtained using very large numbers (PBS: n=72; poly(I:C): n=23; IAV: n=43). Data shown in Figure 2f also contain at least n=6 data points and are in line with findings in Figure 2e. The statistical analysis was performed as indicated.

#3: Sup Fig.2 mentions protective antibodies but this is not shown. Interesting to quantify. Data shown in Fig. 3E-J are not convincing. **A mock control is lacking.** In Fig. 3K (and not Sup 3K), there is a higher bacterial load (6 weeks) but this does not translate into higher pathogenicity. Please explain. Strange way to represent the bacterial loads. **What happens in 2 weeks old offspring (bacterial/viral loads affected too?). In general, do the authors think that after 6 weeks the enhanced susceptibility of offsprings is maintained? Is it really a long-lasting response (line 423)?**

Data shown in Supplementary Figure 2 (now Supplementary Figure 3) is only shown to demonstrate the validity of the established second hit model. Cytokine responses in the offspring shown in Figure 3 e-j are not showing major differences, except may be for IL-2. This is appropriately discussed now. The respective results section is also amended accordingly. Mock controls (PBS treated groups at all conditions shown) were performed but not shown in order to focus the data. We have now mentioned this in the respective results section.

Regarding Figure 3k: yes, we have now performed the requested experiments and measured in addition to MRSA loads also IBV loads in 2-week as well as 6-week old offspring. The results are presented as **new Figure 3 k and l** as well as **new Supplementary Figure 4j and k**. The reviewer is correct, the new data show that the highest vulnerability second hits is in early life (2 week old offspring) and not in adulthood (6 week old offspring). We have amended the respective sections in the manuscript.

#4: In Figure. 4, the number of animals should be increased to enhance statistics. Other progenitors should be investigated. Is there a difference in term of cell number?

In Figure 4, only the Treg data have an n=3. However, all other groups have an n=4-9. With multiple control groups and both sexes assessed, the results are reproducible and statistically robust. Representative flow plots and gating strategies are now shown in the **new Supplementary Figure 9**.

#5: The experiment proposed in Fig. 5 is elegant and should be extremely difficult to accomplish. I however suggest to analyze other criteria (not only the weight). Data from the in vitro experiment are not extremely convincing in terms of differences. It would be interesting to perform RNAseq analysis of alveolar macrophages.

We agree with the reviewer that other additional parameters should be measured. We have therefore determined virus titres after transfer of alveolar macrophages and viral challenge. The new data is now presented as **new Figure 5 h and i**.

Additionally, we measured cytokine responses after adoptive transfer of alveolar macrophages before and after second hit. The new data are now presented in the **new Supplementary Figure 7**.

We agree that further analysis of the transcripts of the respective macrophages would clearly be of interest but contend that it is beyond the scope of this study.

#6: Many links are lacking in this ms. In general, the link between what happens in the BM and what happens in the lungs is lacking. The link between MIA and altered response of offsprings is also not shown. Do the authors think that alteration of the gut microbiota in infected pregnant mice participate in altered response of offsprings (altered microbiota has been shown to impact on hematopoiesis and on lung defence). FTE experiments are realisable. To conclude, more insight should be placed to enhance the quality of this very interesting story.

Potential mechanistic links have been discussed now as requested.

FTE experiments, while interesting, are unfortunately beyond the scope and focus of this study.

Other comments

The authors should mention the literature describing the impact of influenza on hematopoietic stem cells. In general, the text of the Figure legends is by far too long.

We have now carefully revised the discussion including the requested references.

We tried to keep the legends as short but also as informative as possible to allow the reader to judge the data presented without the need to read the M&M section in detail first.

REVIEWER COMMENTS

Reviewer #1 (Remarks to the Author):

The authors have done an excellent job of responding to all of the reviewers' critiques. They have provided additional data which further strengthens their conclusions. I believe it is suitable for publication.

Reviewer #2 (Remarks to the Author):

The authors has largely address my previous concerns.

Reviewer #3 (Remarks to the Author):

The authors have addressed a part, but not the totality, of my comments. I have also other comments regarding new data included in the revised version. In general, there is an overstatement of the results and the conclusions are not convincing.

The authors have not adressed this point : « In panel D, I am surprised that the viral load remains constant at day 6 p.i. ». Usually, the viral load is much lower at day 6-7. What happens after day 6 ?

Infected pregnant mice do not loss weight after infection (Fig. 1B) despite a high viral load in the lungs. Please explain. The level of cytokines induced by p(I :C) is very low (lung and plasma). In my comment « The lack of inflammatory cytokines in p(I :C)-inoculated mice is not surprising», it should be read « IS surprising » (sorry about that). Please explain. Cytokine production in non-pregnant infected mice should be shown as a control (Fig. 1f/m).

The authors have not adressed this point : « Have the authors analyzed the pathology as well (panel m is not commented) ». This analysis may explain the effect on the weigh (no loss in pregnant mice versus loss in non-pregnant mice).

Sup Figure. 2, Figure. 3 : The authors found differences between males and females but this finding is not further exploited.

Figure. 3b/c. A survival curve with only 5-7 mice cannot be conclusive. To be repeated.

Figure 3k is not convincing. The difference is very low and only seen at day 3. At 6 dpi, there is a reduced number of viruses (unlike at 3 dpi). Possible to change the scale to be more convincing ? Difficult to conclude with n=4. To be repeated. It would be better to enhance the n and focus on a single time point p.i. (3 dpi ?).

Lines 257. Yes this is delayed in the case of bacterial infection (panel i) BUT not in the case of IBV (panel k) infection. Here too, the viral load is quite constant at day 1 and 6. It should be reduced. The enhanced susceptibility is only observed in 2 wk old offspring which is an interesting (new information). I guess, regarding the conveyed message, that AM have recovered their function (?).

The authors have not adressed this point. « In Figure. 4, the number of animals should be increased to enhance statistics ». I maintain that n=4 (e.g. for panels b-f) and one experiment performed is not enough to draw a conclusion. Please justify the removal of « outliers ».

The authors have not adressed this point. « Other progenitors should be investigated. Is there a difference in term of cell number ? ». To precise my comment, there are interesting cells downstream including MDPs, CDPs, etc. The authors should make a link between the effects seen on progenitor cells and AM. It would be a good idea to better justify this chapter.

Figure. 5 is not convincing. The % weight change is not the best marker to analyze (and data are not convincing) and the novel panel is not convincing. Where is the PBS control ?

The authors have not adressed this point. « It would be interesting to perform RNAseq analysis of alveolar macrophages ». As the IFN/ISG pathway is critical in anti-viral response, the authors should analyze this pathway in their system.

Fig. 5j, only n=3...

Typo error line 142 « humane » (I don't understand this sentence), line 202 « AIV » should be « IAV »

Point-by-Point Response

The authors have addressed a part, but not the totality, of my comments. I have also other comments regarding new data included in the revised version. In general, there is an overstatement of the results and the conclusions are not convincing.

We thank the reviewer for his/her additional comments and criticism, which we have now addressed as described below.

#1: The authors have not addressed this point : « In panel D, I am surprised that the viral load remains constant at day 6 p.i. ». Usually, the viral load is much lower at day 6-7. What happens after day 6 ?

These data were Figure 1d in the original version and are shown as Figure 1e in the revised version. In the revision, we had included more data points from a new experiment and corrected the pfu/ml to pfu/g, which is more accurate. We apologize for the mistake in the labeling of the y-axis that is now corrected and shows a trend towards lower virus titers at day 6 compared to day 3 p.i.. However, we would like to note that delayed virus clearance is characteristic for the allogenic mouse pregnancy model as we described before (Engels et al., Cell Host & Microbe 2017). It is in line with clinical studies that immunotolerant/immunocompromised patients present prolonged virus shedding. This is represented by the allogenic pregnancy model. In the syngenic mouse pregnancy model, however, as the reviewer correctly assumes, virus titers decline significantly at day 6 p.i. (Engels et al., Cell Host & Microbe 2017). To avoid confusion, we cited the above mentioned study in the respective results section.

#2: Infected pregnant mice do not loss weight after infection (Fig. 1B) despite a high viral load in the lungs. Please explain.

“Weight loss” in the murine pregnancy model is observed in Figure 1b as “delayed weight gain”. This is also in line with our previous publication establishing the murine pregnancy model (Engels et al., Cell Host & Microbe 2017). However, we now include the assessed morbidity scores of the data shown in Figure 1b that show that “delayed weight gain” correlates with higher morbidity scores (Supplemental Figure 1a).

#3: The level of cytokines induced by p(I :C) is very low (lung and plasma). In my comment « The lack of inflammatory cytokines in p(I :C)-inoculated mice is not surprising», it should be read « IS surprising » (sorry about that). Please explain. Cytokine production in non-pregnant infected mice should be shown as a control (Fig. 1f/m).

As stated previously in the manuscript, we used 4 mg/kg poly(I:C) that resulted in plug-to-pregnancy rates comparable to IAV infected dams, while a dose of 20 mg/kg poly(I:C) caused abortion (see Figure, left). In this figure, the plug to pregnancy ratio is presented for multiple experiments (one data point is one experiment).

The number of total dams per group is 71 (PBS,) 51 (poly(I:C), 4 mg/kg), 15 (poly(I:C), 20 mg/kg) and 56 (IAV).

Cytokine induction in the plasma is comparable between the poly(I:C) and IAV groups (Figure 1i-k). In the lung, however, cytokine induction is higher in IAV infected groups than in poly(I:C) treated groups (Figure 1f-h) **as expected**. Upon the request of this reviewer, we had repeated these experiments by applying poly(I:C) intranasally that even proved to be less potent in cytokine induction than the intraperitoneally treated groups (see Supplementary Figure 1c-p). Please also see Figure 2g-i, where we show that the poly(I:C) treatment used in this study is sufficient to cause low birth weight at various time points after birth. Cytokine production in non-pregnant mice is not the focus of this study and was extensively studied and reported in Engels et al., Cell Host & Microbe 2017.

#4: The authors have not addressed this point : « Have the authors analyzed the pathology as well (panel m is not commented) ». This analysis may explain the effect on the weight (no loss in pregnant mice versus loss in non-pregnant mice).

We now describe the pathology in more detail in the respective results section. Please see also response to comment#2.

#5: Sup Figure. 2, Figure. 3: The authors found differences between males and females but this finding is not further exploited.

We assume the author means Supplemental Figure 3. The focus of this experiment is the validation of the pregnancy model used regarding heterologous and homologous challenges as described before. Findings in Figure 3b were further exploited as all experiments were performed in both sexes.

#6: Figure. 3b/c. A survival curve with only 5-7 mice cannot be conclusive. To be repeated.

We have now sought the advice of an expert in statistics who scrutinized all raw data. Experiments shown in Figure 3b and c were performed in three independent experiments, while only one single experiment was shown in the previous version. However, we now merged data of independent experiments resulting in higher animal numbers (see amended figure legend). The results remain unchanged compared to the original version. Statistical analysis confirms findings in Figure 3b and c as solid and reproducible, with multiple events (death of animal) recorded at various time points in Figure 3b (male cohort) but not 3c (female cohort).

#7: Figure 3k is not convincing. The difference is very low and only seen at day 3. At 6 dpi, there is a reduced number of viruses (unlike at 3 dpi). Possible to change the scale to be more convincing ? Difficult to conclude with n=4. To be repeated. It would be better to enhance the n and focus on a single time point p.i. (3 dpi ?).

We discussed these data with our statistical expert and agree that there is only a difference at day 3 p.i.. However, since we used a log scale for visualization purposes, the present difference was difficult to see by the naked eye. Therefore, we now show data in Figure 3k

and l as fold-changes in comparison to the average PBS signal at each respective time point p.i.. It is now better visible that 3 out of 4 animals born to IAV infected dams show 3-12 times higher IBV loads. We do not agree that we should show only the time points where differences are detected since it is also important for the overall conclusions to see the kinetic of virus clearance. Additionally, we now show the original data (previously shown in Figure 3k and l) as Supplementary Figure 6 to allow the reader full access to the raw data. However, we adjusted our conclusions regarding this data carefully in the respective results section in order to avoid overstatements.

#8: Lines 257. Yes this is delayed in the case of bacterial infection (panel i) BUT not in the case of IBV (panel k) infection. Here too, the viral load is quite constant at day 1 and 6. It should be reduced.

We have now amended the respective conclusion in the results section. Please also see responses to comment#1 and #7.

#9: The enhanced susceptibility is only observed in 2 wk old offspring which is an interesting (new information). I guess, regarding the conveyed message, that AM have recovered their function (?).

Immune cell frequencies of 6-week old offspring are shown in Supplementary Figure 6. We cannot conclude on function but it seems that their frequency is not changed anymore in 6-week-old offspring compared to 2-week-old offspring.

#10: The authors have not addressed this point. « In Figure. 4, the number of animals should be increased to enhance statistics ». I maintain that n=4 (e.g. for panels b-f) and one experiment performed is not enough to draw a conclusion. Please justify the removal of « outliers ».

Data shown in Figure 4 were statistically scrutinized by an expert in statistics and revealed that IAV infection is the major confounder in this data set, while the sex of the animals was not identified as a potential confounder. Thus, we pooled female and male data sets as we had done before in other figures because no sex-dependent difference was observed. Subsequently, statistics were repeated with increased statistical power, which reproduced our initial results and conclusions and allowed a clear presentation of the data. Definition of outliers is now available in the respective Materials and Methods section.

#11: The authors have not addressed this point. « Other progenitors should be investigated. Is there a difference in term of cell number ? ». To precise my comment, there are interesting cells downstream including MDPs, CDPs, etc. The authors should make a link between the effects seen on progenitor cells and AM. It would be a good idea to better justify this chapter.

We agree that further down-stream experiments would be interesting but contend that this is beyond the scope of the present study. We did not measure absolute cell numbers but instead compared immune cell populations according to previously established protocols for the allogenic pregnancy model (Engels et al., Cell Host & Microbe 2017). We find that absolute

numbers are dependent on recovery of cells which is quite variable, whereas relative numbers and ratios are more consistent. Links between progenitor cells and alveolar macrophages were reported before and discussed in the last revision accordingly.

#12: Figure. 5 is not convincing. The % weight change is not the best marker to analyze (and data are not convincing) and the novel panel is not convincing. Where is the PBS control ?

We disagree with this comment. Weight loss as a representative for “clinical outcome” is the strongest parameter to be detected. Since the data shown are statically robust, there is no objective evidence to doubt the data. We did initially not show PBS data (which are all negative as expected) in order to provide clear data presentation without unnecessarily complicating the figure. We now include this information in the respective figure legend.

#13: The authors have not adressed this point. « It would be interesting to perform RNAseq analysis of alveolar macrophages ». As the IFN/ISG pathway is critical in anti-viral response, the authors should analyze this pathway in their system.

We agree that RNAseq experiments on immune subsets would be very interesting. However, this is unfortunately beyond the scope of this study.

#14: Fig. 5j, only n=3...

We agree that these data are statistically less robust. Since these data were re-capitulating in vitro data sets shown robustly in vivo, we decided to remove these data from the manuscript.

#15: Typo error line 142 « humane » (I don't understand this sentence), line 202 « AIV » should be « IAV »

We corrected “AIV” to “IAV” in the stated line. The implementation of “humane endpoints” is a key component of refining studies involving laboratory animals. The use of humane endpoints in animal experiments describes the identification of clear, predictable and irreversible criteria which substitute for more severe experimental outcomes such as death.